# Targeted analysis of dyslexia-associated regions on chromosomes 6, 12 and 15 in large multigenerational cohorts

**Nicola H. Chapman[1], Patrick A. Navas[1], Michael O. Dorschner[1], Michele Mehaffey[2¤a], Karen G. Wigg[3], Kaitlyn M. Price[3,4,5], Oxana Y. Naumova[6], Elizabeth N. Kerr[4,5], Sharon L. Guger[4], Maureen W. Lovett[3,5], Elena L. Grigorenko[6,7], Virginia Berninger[8¤b], Cathy L. Barr[3,8,9,10,11,12], Ellen M. Wijsman** [1,13¤b]*, **Wendy H. Raskind** [1,14¤b]*

1 Division of Medical Genetics, Department of Medicine, University of Washington, Seattle, Washington, United States of America, 2 Department of Pediatrics, Division of Pediatric Genetics, Pediatric Genetics, University of Washington, Seattle, Washington, United States of America, 3 Program in Neuroscience and Mental Health, Hospital for Sick Children, Toronto, Ontario, Canada, 4 Department of Psychology, Hospital for Sick Children, Toronto, Ontario, Canada, 5 Department of Pediatrics, University of Toronto, Toronto, Ontario, Canada, 6 Department of Psychology, University of Houston, Houston, Texas, United States of America, 7 Department of Molecular and Human Genetics, Baylor College of Medicine, Houston, United States of America, 8 Department of Educational Psychology, University of Washington, Seattle, Washington, United States of America, 9 Division of Experimental and Translational Neuroscience, Krembil Research Institute, University Health Network, Toronto, Ontario, Canada, 10 Department of Physiology, University of Toronto, Toronto, Ontario, Canada, 11 Institute of Medical Sciences, Toronto, Ontario, Canada, 12 Department of Psychiatry, University of Toronto, Toronto, Ontario, Canada, 13 Department of Biostatistics, University of Washington, Seattle, Washington, United States of America, 14 Department of Psychiatry & Behavioral Sciences, University of Washington, Seattle, Washington, United States of America

¤a Current address: Pacific Northwest Research Institute. 720 Broadway, Seattle, WA 98122
¤b Professor Emerita
* wendyrun@uw.edu (WHR); wijsman@uw.edu (EMW)

## Abstract

Dyslexia is a common learning impairment with a genetic basis that affects word reading and spelling. An increasing list of loci and genes have been implicated, but analyses to-date have investigated only limited genomic variation within each locus with no confirmed pathogenic variants identified. Our study is the first to comprehensively sequence both coding and cis-acting regulatory regions of such genes in a large study sample. In a collection of >2000 participants in families from three independent sites, we performed targeted capture and comprehensive sequencing of all exons and some regulatory elements of five candidate risk genes (*DNAAF4*, *CYP19A1*, *DCDC2*, *KIAA0319* and *GRIN2B*) for which prior evidence for a role in dyslexia exists from more than one sample. We evaluated evidence for association in each of six dyslexia-related quantitative phenotypes (traits) using both individual common single nucleotide polymorphisms and aggregated rare variants. We detected no promoter alterations and few deleterious variants in the coding exons, none of which showed evidence of association with any trait. Single variant and aggregate testing of *DNAAF4* failed to detect significant evidence of association with any of the

**Data availability statement:** All relevant data are within the manuscript and its Supporting Information files.

**Funding:** Support was provided in part by grants from the Eunice Kennedy Shriver National Institute of Child Health and Development (https://www.nichd.nih. gov/) 1R01HD088431 to WHR and EMW, P50HD33812 to CLB, and P50HD05212 (Project 6) to ELG, grants from the Canadian Institutes of Health Research (MOP-133440 and PJT-180419) to CLB (https://cihr-irsc. gc.ca/e/193.html). K.P. was supported by the Hospital for Sick Children Research Training Program (Restracomp; https://www.sickkids. ca/en/research/research-training-centre/ scholarships-fellowshipsawards/). The funders had no role in study design, data collection and analysis, decision to publish, or preparation of the manuscript.

**Competing interests:** The authors have declared that no competing interests exist.

traits. The other four genes provided evidence of association with one or more traits. A common variant downstream of *CYP19A1* showed significant evidence of association with multiple traits with or without verbal IQ (VIQ) adjustment. A haplotype that stretches from the downstream region of *KIAA0319* to the second intron of *DCDC2* was associated with reduced performance on timed real word reading. Finally, rare exonic variants in *GRIN2B* were associated with performance on spelling, with or without adjustment for VIQ. Our findings from this large-scale sequencing study complement those from genome-wide association studies, argue against the causative involvement of large-effect coding variants in these five candidate genes, support a multigenic etiology, and suggest a role of transcriptional regulation.

## Introduction

Dyslexia is a complex learning impairment of neurobiological origin that can be defined as unexpectedly low accuracy and/or rate of oral reading of single words or pronounceable pseudowords, or low accuracy of spelling [1]. It manifests as difficulty in learning to read and spell despite adequate instruction and is not attributable to general cognitive impairment, primary sensory or motor impairment, psychiatric or other neurologic disorder or delays in aural or oral language. The estimated prevalence of dyslexia varies depending on ascertainment schemes, exclusion criteria, tests included in diagnostic assessment, and thresholds used for a categorical diagnosis. In school-aged children, most estimates of dyslexia fall between 5–12% [2–5] but have been as low as 3.5% [6] and as high as 20% [7] In almost all past studies, including our own, males are at greater risk than females for both presence of dyslexia and its severity [2,5,8–10]. Even with educational intervention, many aspects of dyslexia can persist into adulthood, including slow reading speed and poor spelling-related writing abilities [11–13] leaving lasting impacts on self-esteem, educational opportunities, and occupational choices [14–16].

Multiple lines of evidence, including twin [17], familial aggregation [8,18], adoption [19], and linkage and/or association studies [20], have led to the consensus that there is a substantial genetic contribution to dyslexia and its component phenotypes. Heritability estimates are as high as 50–70% [21,22]. Although rare families have been described in which dyslexia appears to be transmitted as a single gene disorder [23–28] studies in the general population show that, like most complex traits, dyslexia and its correlated underlying processes are genetically heterogeneous and likely involve the influence of variation in multiple genes [29]. Such heterogeneity complicates identification of underlying genes, regardless of the study design, but multiple candidate susceptibility genes have been nominated from genomic regions of interest (ROIs) identified by linkage analyses [30,31], genome wide association studies (GWAS [32–34]), copy number scan [35], structural chromosome rearrangements [36,37], or whole genome sequencing [27].

Of the many reported ROIs for dyslexia risk, a small number have received support by more than one research group on independent samples. The most

prominent are DYX1 on chromosome 15q [38–44] and DYX2 on chromosome 6p [39,45–48]. Further analyses of these regions identified a small number of candidate genes. In particular, dynein axonemal assembly factor 4 (*DNAAF4*, MIM:608706) and cytochrome P450 family 19 subfamily A member 1 (*CYP19A1*, MIM:613546) [37,49,50] in DYX1 and double cortin domain containing 2 (*DCDC2*, MIM:605755) and *KIAA0319* (MIM:609269) in DYX2 are the candidate genes that have been the most investigated [51–59]. Our linkage analyses in the University of Washington (UW) cohort for various quantitative measures used to assess dyslexia identified additional candidate loci [60–62]. In the UW sample one of the strongest linkage signals was in a region on chromosome 12p [61]. This region contains glutamate receptor, ionotropic, N-methyl-D-aspartate 2B (*GRIN2B*, MIM:13249), a gene that had support as a dyslexia candidate gene from studies in other data sets [63–65].

While support for involvement of the aforementioned genes has been reported from both a variety of association and linkage analyses and functional studies, evidence favoring particular genes in the ROIs is inconsistent or difficult to interpret [66–72]. There have been failures to detect linkage [73–75] or association [76–82], as well as reports of increased risk attributed to opposite alleles [76,83,84]. For a complex trait there is also the chance that composite/synthetic quantitative trait loci (QTLs) are responsible for some of the linkage analysis results [85,86]. False-positive results are another possible explanation. Demonstration of potential functional competence of the putative risk allele in an animal model is also difficult to interpret in the context of a human trait [87]. Meta-analyses have not resolved these conflicts [80,88–90], nor have modest-sized GWAS, which have provided at most weak support for the loci [34,91–96]. This is also the case for a recent large GWAS that failed to detect significant evidence of association with any reported candidate dyslexia risk gene [32]. However, the large sample size was only feasible though use of cases without a clinical diagnosis. This is a situation that can lead to statistical heterogeneity in results, raising concerns about usefulness of such samples, as has been reported in application to another complex trait [97]. A recent highly-targeted sequencing study [98] of specific learning disorders noted the existence of an exome variant in *KIAA0319*, but the small sample size (37 people) limited power to achieve statistical significance. Variability in conclusions across the different study designs and samples is common and not surprising. Genetic heterogeneity has been responsible for discrepant results since the earliest days of genome scans, even for "simple" Mendelian traits [99] and genome-wide linkage analyses and GWAS both allow location scans, but with different sensitivities to less vs. more-common trait-gene allele frequencies [100], and with power to detect genetic effects influenced by sample ascertainment procedures [101]. Neither approach queries all the genes or DNA variation, which requires more-expensive DNA sequencing of at least the regions of interest.

The putative effect of candidate genes on neuronal migration has been used to bolster their credibility [102,103], given early reports of cortical brain abnormalities in people who were thought to have had dyslexia [104,105]. However, although cortical abnormalities have been observed with knockdown of the rat orthologs *Dnaaf4* [87], *Kiaa0319* [66], or *Dcdc2* [106], this is not observed in knockout mice [107–109], and the cortical migration hypothesis remains unproven [69]. Observations that dyslexia candidate genes seem to have a role in ciliogenesis [110,111], synaptic transmission [112], or axonal growth [113], have led to alternative hypotheses of pathogenesis.

Although issues described above are to be expected in a complex disorder, to date no causative pathogenic variants have been confirmed for dyslexia or quantitative traits used in its diagnosis. Some possible explanations for this failure include: (1) genetic and/or phenotypic heterogeneity that masks detection in samples ascertained and phenotyped with different criteria; (2) risk element(s) may alter expression of the protein but not its amino acid sequence; (3) risk elements may escape recognition but affect splicing; and (4) the number of samples sequenced comprehensively has been too small to have the power to detect variants of modest effect size [98].

Recent advancements in DNA sequencing methods now enable the larger scale sequencing efforts that are necessary to evaluate genetic variation in ROIs more comprehensively than was possible earlier. This technology allowed us, in a multi-site study reported here, to investigate the potential role of variants of smaller effect size, non-coding variants, and sample heterogeneity as explanations for previous variable results in ROIs implicated in dyslexia. To search

for variants that show evidence of association with dyslexia-related traits, we report, here, the results of genomic sequencing and association analyses in a collection of >2000 participants in families with members who have dyslexia and shared phenotypic measures enrolled at three institutions. We report results from a comprehensive analysis of the coding regions and some regulatory element motifs of five putative dyslexia risk genes to assess their possible role in performance on six tasks that yield quantitative scores and are commonly used in the evaluation for dyslexia. The analyses focused on two highly cited loci and a genomic region implicated by our previous studies and supported by the literature. We present additional evidence for a role in dyslexia risk for *DCDC2*, *KIAA0319*, *GRIN2B* and *CYP19A1*, but not for *DNAAF4*.

## Materials and methods

### Overview of rationale and data used

We targeted a limited portion of the genome for deep investigation. The comprehensive high-throughput sequencing approach used here allowed a relatively complete investigation of association of DNA variation with dyslexia-related traits in a large sample. The use of sequence data provides potential to identify causal nucleotides rather than only localizations. Practical issues of number of genes investigated were driven by cost, sample size, and challenges of interpreting genomic sequence data in non-coding sequence. To maximize sample size, we sequenced every individual in our combined dataset who had the relevant phenotypic data. This strategy gave us capacity to evaluate five genes and regulatory/splice regions around those genes in three genomic regions.

The loci DYX1 and DYX2 have the greatest support across independent samples with hundreds of citations since initial reports [114,115] These loci were initially proposed through linkage analyses [43,45,116] and are supported by additional linkage studies (*e.g.*, [38,39,46,74,117,118]. These initial and follow-up analyses used discrete and/or quantitative measures of dyslexia, including reading or spelling-related traits commonly assessed in diagnosis of dyslexia. However, some of these studies had modest sample sizes, not all samples provided strong statistical support, and none carried out comprehensive analysis of DNA around each gene. Therefore, our strategy was to carry out an analysis in a large independent sample of interpretable DNA variants in and near each gene. We focused on four genes in these two regions as both an attempt to replicate previous results in our sample, and to try to identify causal nucleotides. In these two regions, we selected genes *DNAAF4* and *CYP19A1* in DYX1, and *KIAA0319* and *DCDC2* in DYX2.

More details regarding rationale for followup of DYX1 and DYX2 are as follows. In DYX1, the candidate gene *DNAAF4* was first identified via a balanced translocation that segregated with dyslexia in a family [119] and was subsequently supported by family-based transmission studies [40,76,83,84]. Another candidate gene in DYX1, *CYP19A1* [37,52] codes for aromatase, an enzyme that converts androgens to estrogens in the brain [53]. This gene is of interest because of the almost universally observed skewed ratio of males:females with dyslexia. Two other candidate genes in DYX1, phospholipase Cb2 and phospholipase A2 group IVB, were not confirmed in a family-based study [120] and were therefore not investigated here. In DYX2, follow up of the original report of linkage with common SNPs [51–54,106,121] implicates variants in or near *KIAA0319* and *DCDC2*, possibly on a haplotype, in association analyses with dyslexia or reading-related traits. Complementary work *in vitro* and in embryological and rat brain samples provides evidence of altered expression in brain regions believed to be important in reading [66,106].

The 12p region was a novel region identified by linkage analysis in the UW cohort [61]. This region was selected for two reasons. First, it provided one of the strongest significant results and second, because it was obtained with trait data we were using for this study of DYX1 and DYX2. A logical next step is to search for potential sequence variants that might explain the linkage analysis finding. Importantly, this signal was obtained for traits that had also been assessed in the other two cohorts included in the analyses reported herein. In the chromosome 12p region, *GRIN2B* was the only gene that already had some published support to include it as a candidate gene [63–65].

## Participants and phenotypes across sites

**Sample and phenotype selection strategy.** We selected participants enrolled in studies of dyslexia and related phenotypes at three institutions that had overlapping phenotype batteries: University of Washington (UW), The Hospital for Sick Children (SickKids; SK), and the University of Houston (UH). We only used the quantitative phenotypes and not the dyslexia diagnostic status for our analyses; henceforth, we use "traits" to refer to these quantitative measures. We provide here a summary of the sample selections. Extensive descriptions of the UW and SK cohorts have been published previously [8,11,122,123]. Traits were measured by standardized normed tests administered by more than one site and were collected on the original probands, their siblings, and additional family members including parents and sometimes other relatives. This strategy provided a large total number of participants screened with an essentially equivalent test-battery allowing for joint analysis across the three cohorts while minimizing introduction of excess phenotypic heterogeneity that may be introduced by mixing different phenotypic measures. Even so, some variability in underlying risk allele frequencies is expected across cohorts here because details of recruitment invariably differ across recruitment sites, as is the case for virtually all analyses that aggregate data from multiple sites. This leads to biased estimates of risk-allele effects [124] but does not affect interpretation of the results of hypothesis testing results. The sample is largely separate from other cohorts analyzed for association of dyslexia with the ROIs investigated in the current study, thus providing an independent evaluation. Under the assumption that genetic heterogeneity in dyslexia may be reflected in phenotypic heterogeneity, the focus was on individual subtests or index scores based on multiple subtests in standardized, nationally normed tests that are predictive of reading or spelling outcomes or that assess processes related to reading and spelling achievement such as verbal reasoning or phonological memory. To maximize sample size, only test measures administered by more than one of the three cohort sites described below were evaluated. For all cohorts, individuals with evidence of intellectual disabilities, neurological or severe psychiatric disorder, or known genetic disorder associated with language impairment were excluded. Children and related adults with both trait and genotype data were included in the analyses.

**University of Washington (UW) Cohort.** Recruitment and evaluation of probands and their multigenerational family members were done under a protocol approved by the University of Washington Institutional Review Board, are comprehensively described elsewhere [8,11], briefly summarized herein, and provided in more detail in the Supporting Information document. For the UW cohort a discrepancy criterion was used for qualification of a child as a proband under a model of specificity of the trait. Probands who qualified their families for participation had to have a prorated verbal IQ (VIQ) ≥ 90 (≥25%ile) on the *Wechsler Intelligence Scale for Children—3rd Edition (WISC-3)* [125], and score below the population mean and at least 1 standard deviation below their VIQ on at least two measures of accuracy or rate of single real or nonword reading or accuracy of spelling from dictation. as assessed by the nontimed Word Identification (WID) and Word Attack (WA) subtests of Woodcock Reading Mastery Test-Revised (WRMT-R [126]), spelling subtest of the Wide Range Achievement III (WRAT-III; [127], and timed Sight Word Efficiency (SWE) and Pseudoword Decoding Efficiency (PDE) subtests of the Test of Word Reading Efficiency (TOWRE; [128]). Written informed consent and/or assent was obtained from all participants. Ascertainment of subjects for this project began on 9/11/1996 and ended 8/9/2005. The full data set consists of 2079 individuals in 284 families. Of the available subjects, 1347 individuals from 278 families provided quality DNA samples for sequencing. Phenotypic data were available for 96.8% of the 1333 samples that passed quality control (QC) testing. Family sizes ranged from 3 to 51 individuals, with a median family size of 12 in 2–4 generations. Self-reported ethnicities were as follows: non-Hispanic White (90%), Asian (2.1%), Native American (2.0%), African American (1.1%), Hispanic (0.8%), and Pacific Islander (0.2%).

**The SickKids (SK) Cohort.** Details of the ascertainment, assessment, and inclusion/exclusion criteria for the SK cohort have been comprehensively described previously [122,123], are briefly summarized herein, and are provided in greater detail in the Supporting Information document. Probands were children aged 6–16 in schools in the greater Toronto area and Southern Ontario, with WISC-3 or WISC-4 Verbal and Performance IQ [125,129] ≥80 and a score at

least 1.5 SD below the mean on 2 of 3 measures of single real- or non-word reading, or 1 SD below the mean on all 3. Assessments included subtests from the WRAT-III, TOWRE and WRMT-R. Written informed consent and/or assent was obtained from all participants under protocols approved by the Hospital for Sick Children and University Health Network Research Ethics Boards. Families were recruited for genetic studies from 11/23/1999–10/2/2017. The SK cohort comprised 816 participants from 245 families (155 with the proband and one or both parents and 90 with two or more children and one or both parents). Phenotypes were available for 99% of children (349 of 351), and 85% of parents (394 of 465). The phenotype battery in parents was limited to TOWRE SWE and PDE subtests [128]. Self-reported ancestry was available for 185 (76%) of the families. Of these, 180 (73.5%) reported European or European-Canadian ancestry. The remaining families reported small amounts of indigenous ancestry (3 families, 1.6%), African ancestry (1 individual) and Mexican ancestry (1 individual).

**The University of Houston (UH) Cohort.** The subset of families of probands with dyslexia used here is a component of the ongoing collection supported by the Florida Learning Disabilities Research Center (FLDRC). Written informed consent was obtained from all participants under protocols approved by the Research Ethics Board of the University of Houston. Participants in this study were recruited between 11/01/2017 and 12/30/2018. The probands were children aged 8–17 who had problems reading and were native English speakers. They were registered with FLDRC and recruited through their registration. The probands and their siblings in the same age range were administered a battery of tests including IQ and reading abilities. Designation of affected status required the Wechsler Abbreviated Scale of Intelligence (WASI [130]), score ≥80 and a score at least 1.5 SD below the mean on 2 of 3 measures of single real- or non-word reading (WID and WA from the WRMT-R, SWE and PDE from the TOWRE), or 1 SD below the mean on all 3 indicators. Self-reported ethnicities were as follows: African American (13.5%) and Hispanic (86.5%). This cohort consisted of 49 participants (37 children and 12 parents) from 15 nuclear families. Parents provided blood samples but were not phenotyped.

## Molecular methods

**Overview of approach.** The targeted capture approach used here was designed to produce a comprehensive assessment of sequence variants relevant to the subset of dyslexia candidate genes evaluated. In the large, combined sample, this included all coding variants, as well as non-coding variants potentially involved in some aspects of gene regulation. For this investigation, the focus was on the potential impact on RNA processing and/or transcription-factor (TF) motifs in tissue-dependent open chromatin regions that control gene expression.

**Genomic regions investigated.** We targeted DNA sequence in three previously published regions of interest (ROIs) for comprehensive evaluation. As described in the Introduction, two of the ROIs have been widely investigated. These two loci, DYX1 on chromosome 15q and DYX2 on chromosome 6p, have the greatest number of studies and independent samples with support for the region [114] with both positive linkage and association analyses reported by more than one group: the DYX1 locus on chromosome 15q and the DYX2 locus on chromosome 6p. Previous linkage analyses in the UW cohort provided support for DYX1 as a dyslexia candidate region [74,83] and similar studies in subsets of the SK cohort supported both DYX1 and DYX2 [122,131,132]. From each of these ROIs, we selected for study two genes based on prior publications supporting a role in dyslexia: in DYX1, *DNAAF4* [40,44,83,84,119] and *CYP19A1* [37,50,133]; and in DYX2, *KIAA0319* [52,66,81,134,135] and *DCDC2* [89,121,136]. Intron 2 of *DCDC2* [106] contains READ1 – a complex compound repeat polymorphism that was previously proposed as the functional dyslexia-risk component in the DYX2 region [137–139] and was included as part of our capture-design. We also selected the glutamate ionotropic receptor NMDA type subunit 2B *(GRIN2B)*, in a third ROI, on chromosome 12p. This region was among those with the strongest evidence of linkage from the UW family studies, with support across several test-battery items [61,140,141]. *GRIN2B* has also been implicated as a dyslexia risk factor [63–65]. For analysis of these five genes, we developed a set of custom capture probes to enable comprehensive evaluation of potential regulatory and splice region sequence variants in addition to coding region variants, as described below.

**Sample preparation.** For most samples, genomic DNA was extracted from peripheral blood mononuclear cells or Epstein–Barr virus-transformed B-lymphoblastoid cell lines. When only saliva samples were available, DNA was extracted using a DNAGenotek OGR-500 kit (DNAGenotek Inc, Ontario, Canada) according to the manufacturer's instructions.

**Single molecule molecular inversion probe (smMIP) targeted capture and sequencing.** The capture target consisted of all potentially functional sequences and variants in each ROI around and including each of the five selected genes. This included a total of 277 potential regulatory regions. We used smMIPs to capture targeted DNA with methods described elsewhere [142,143], followed by multiplex sequencing. Additional details are provided in the Supporting Information document.

We used the UW pipeline [144] to design smMIPs to capture all exons and 10–20 bp of flanking intron sequences. This approach ensured capture of the splice site branch A points (RefSeq, hg19/GrCH37 build [143]). All analyses were on this genome build. For the non-coding regulatory regions, we used the ATAC-seq data in Brain Open Chromatin Atlas (BOCA) [145,146] and ENCODE Consortium for the brain specific (including fetal brain) DNaseI hypersensitive sites [147] to identify chromatin accessible regions (CARs) from 80 kilobases (kb) upstream of the transcription start site (TSS) to the same distance downstream of the 3´ untranslated region (3´ UTR) in the genes of interest. S1a Table lists the targeted regions and their annotations. The smMIPs were designed to minimally overlap each ~200 bp DHS or ATAC-seq site with an additional 50 bp of flanking sequences. The resulting 1574 smMIPs and a 55 probe smMIP fingerprinting collection were pooled, tested on a set of control DNAs, and rebalanced, resulting in a final pool of 1569 smMIPs (Supporting Information - Methods and S1b Table).

**Multiplexed next generation sequencing.** Libraries were prepared [142,148] and pooled for sequencing in batches of 384. Each pool was sequenced using standard paired-end (100 bp) rapid run chemistry in a single lane on a HiSeq 2500 (Illumina, San Diego, CA). The final batch contained repeats from previous batches. Using a quality control (QC) benchmark requiring that each sample have a minimum of 80% of target bases covered with a depth of at least 10, 2040 (92%), 2190 (99%), and 2176 (98%) of the 2209 samples prepared passed the benchmark on chromosomes 6, 12, and 15, respectively. For de-multiplexing, generation of FASTQ files, and annotation of sequence data, we used the same in-house pipeline as for MIP design (see Supporting Information). We called a total of 2026 variants – 341 in *DCDC2*, 376 in *KIAA0319*, 685 in *GRIN2B*, 511 in *CYP19A1* and 113 in *DNAAF4*. After QC steps, the sample sizes were 1333, 782 and 46 for the UW, SK and UH samples, respectively, with average genotype completion rates of 98.7%, 99.2% and 98.9%, respectively. There were 297 variants remaining in *DCDC2*, 330 in *KIAA0319*, 654 in *GRIN2B*, 496 in *CYP19A1* and 100 in *DNAAF4*. Variants were annotated using the 1000 Genomes Project (1KGP) and Ensembl Variant Effect Predictor (VEP) [149].

## Statistical and bioinformatic analyses

**Overview of analysis approaches.** We carried out a comprehensive association analysis of performance on six tasks commonly used in the evaluation for dyslexia. We did not analyze dyslexia *per se*. We employed a standard family-based design, used widely for studies of traits with a genetic basis. This design uses both impaired and unimpaired individuals in the analysis. The (also standard) analysis approach that we used seeks evidence of concordance of genotypes or alleles among individuals with similar phenotypes, and discordance between individuals with different phenotypes. We focused our analysis on a set of participants that included probands with reading difficulties as well their biological relatives with and without reading difficulties. In contrast to many papers where dyslexia is considered as a categorical diagnosis, we used continuous trait data of the six reading-related phenotypes.

The association analyses were done for the quantitative traits and all variants identified by our assays and samples that passed QC. As a first high-throughput sequencing project in this area, we focused only on already nominated genes and surrounding potentially regulatory DNA. Data handling used R packages GWASTools [150], SeqVarTools [151], and GenomicRanges [152] from Bioconductor v3.12 [153]. Association analyses were carried out with GENESIS [152] in

Bioconductor v3.12 [145]. A full range of variant frequencies was considered. Variants with sample frequency greater than 1% were tested individually, and rarer variants were combined in aggregate testing.

**Ancestry adjustment.** Self-reported continental ancestry was available for most samples. Because almost all samples were of European origin, either by self-report or KING ancestry estimation (Supporting Information), a simple European/non-European indicator assigned by self-report was used to adjust for ancestry in all analyses as a potential nuisance covariate. SNP-based ancestry estimates conflicted with self-reported ancestry in only six individuals. SNP-based ancestry was used in these individuals because self-reported ancestry may reflect cultural affiliation rather than genetic ancestry [154].

**Phenotypes and adjustments.** As with all complex traits for which there is heterogeneity across collection sites, this additional heterogeneity adds a cost to the sample size required to detect association by reducing variant effect size in the full sample. However, the only way to achieve sufficiently large samples to detect association with complex traits is to include as many existing sample sets as possible that have assessed the same traits. Reading-related phenotypes used for our analyses that were directly comparable across the three data sets included word identification WID and WA from the WRMT [126], and SWE and PDE from the TOWRE [128]. The UW and SK cohorts also included spelling (SP) from the Wide Range Achievement Test – Revised (WRAT3-R [127]), and nonword repetition (NWR) from the Comprehensive Test of Phonological Processing [155]. For all traits considered here, lower scores indicate more impairment on the measure. In addition, only the UW and SK cohorts included VIQ, which was used as a covariate for some analyses.

Two different phenotype adjustments were considered with a linear model. The first model (UNADJ) included three covariates for non-European ancestry, age and sex only. The second model (VIQADJ included these three covariates and added a covariate for VIQ. The residuals for the first vs second set of adjustments represent traits that can be interpreted as including vs free from VIQ effects. The second set of adjustments could only be performed on the UW sample and the children in the SK sample because of the availability of VIQ and, therefore, has a reduced sample size relative to the UNADJ residuals. Previous analyses in the UW data set [8,60,61,74,156] indicate that these models are appropriate for these phenotypes. Information about socio-economic status or other environmental covariates was not available in any of the three data sets. For brevity, when discussing results, we use the format Trait:Adjustment (*e.g.*, SWE:UNADJ and SWE:ADJ) to refer to the phenotype without or with VIQ adjustment.

**Association testing.** Analysis was done in two phases. In phase 1, a set of covariate-adjusted traits were obtained within each data set (described above), captured as the residuals from the trait-adjustment model. The difference between the UNADJ and VIQADJ analyses comes from the different sets of residuals from the phase 1 analysis. In phase 2, these residuals from phase 1 were jointly used as the response variable in the across-study association testing. In the phase 2 association testing, only the relationship between the individual SNPs and the residuals from phase 1 is of interest, reported, and tested. For all such SNPs observed in two or more copies in the combined data set, we regressed the phase 1 residuals for the trait of interest against the dose of the minor allele, yielding a single-variant test. This overall two-stage approach includes within-study linear covariate effects in phase 1 to provide basic adjustments and also VIQ when relevant. Phase 1 adjustments were done within data sets both because different editions of tests were used across data sets and because there were differences in ascertainment. The phase 2 cross study analysis employed a model that allowed for global residual site-effects that might reflect differential effects of recruitment or other sample features across site in the association testing. Association testing in phase 2 was done on the combined data sets using GENESIS [157] in Bioconductor v3.12 [152], with distinct means and variances modeled to capture residual site effects. Estimating different residual variances in each data set allows joint analysis of data sets without violating the assumption of homoscedasticity that is essential to linear regression. Family relationships were accounted for by using the expected pedigree-defined kinship in the covariance matrix via a mixed model. For all SNPs observed in two or more copies in the combined data set, we regressed the phenotype of interest against the dose of the minor allele, yielding a single-variant test. Because SNPs with minor allele frequency (MAF) > 0.01 should result in approximately 40 copies in our dataset, we conservatively

chose this value as the MAF above which we consider the results of single-variant tests. This assures that test statistics should be robust to allele frequency.

Rare SNPs with MAF ≤ 0.01 were included in aggregate analyses, with grouping according to their location relative to each candidate gene, defined as 5′ region, 3′ region, exons, and introns. The SKAT-O [158] aggregate test was performed in GENESIS, using weights following a Beta distribution with parameters (1,25) and dependent on the MAF [158]. This choice of weight distribution more heavily weights the rarest variants, but still allows for a contribution from more common variants. The SKAT-O test optimizes power by finding the maximal weighted average of the Burden test (more powerful when most variants are causal and effects are in the same direction) and the SKAT test (more powerful when most variants are not causal and effects can be in either direction) and is therefore the best choice in this situation where we do not have an *a priori* expectation of the direction or size of variant effects.

We determined significance thresholds for statistical testing as follows. A significance threshold for single-variant tests must account for the effects of linkage disequilibrium (LD) blocks (*e.g.*, [159]). Such thresholds do not change with increasing marker density [160], but do depend on the population involved, due to differences in LD between populations. A study using 1KGP Phase 3 data showed that in European samples a genomewide threshold of $9.26 \times 10^{-8}$ is most appropriate for single-variant tests with a target type I error rate of 0.05 [161]. We used this genomewide threshold and scaled it to account for the approximate fraction of the genome under analysis. The present study involved 5 genes, compared to approximately 25000 in a full GWAS, so we use $p < 4.63 \times 10^{-4}$ ($25000/5 \times 9.26 \times 10^{-8}$) as a stringent significance cutoff for single-variant tests. This allows for both the number of genes evaluated, and the presence of LD blocks in those gene regions. For aggregate testing, since the gene-region is the unit of analysis independent of presence/absence of LD blocks in the gene-region, further adjustment for the number of LD-blocks is not warranted. To achieve a type I error rate of 0.05, we therefore used a *p*-value of 0.0025 as the cutoff for aggregate tests, motivated by dividing 0.05 by 20 for a simple Bonferroni correction using the number of gene-region tests performed (4 tests for each of 5 genes). We did not adjust test thresholds for analysis of multiple traits because current studies do not typically do so. The limited literature to date shows no evidence for an increased false-positive rate, with the advantages of using multivariate and/or pleiotropic models falling primarily on the side of potential increase of power to detect true, but weak, associations [162].

**Haplotype estimation and testing.** When multiple SNPs in LD with one another achieved significance, we used Beagle 5.4 [163] with the 1KGP European reference population (EUR) [164] to obtain phased genotypes, thus providing pairs of phased haplotypes for each subject. For a locus with *n* common haplotypes, we fit *n* additive models for each phenotype, where the *i*th model estimates the dose effect of haplotype *i* relative to the other haplotypes pooled. GENESIS allowed us to correct for relationships by using the pedigree-defined kinship in the covariance matrix of a mixed model.

**Annotating non-coding variants in CARs.** We explored the potential impact of all non-coding variants with MAF > 0.01 and significant evidence of association with at least one of the UNADJ and VIQADJ phenotypes. We annotated variants using the JASPAR tracks on the UCSC Genome Browser [165] as well as the JASPAR database [166]. We considered four characteristics of non-coding variants that together are suggestive of a regulatory effect: 1) the variant is in the peak signal (~200 bp) in either ATAC-seq [146] or DNaseI-seq brain profiles [167] in any brain region; 2) it overlaps with a known TF motif, as found in JASPAR [166]; 3) the change disrupts a conserved position in the motif, as assessed by the position frequency matrices in JASPAR; and 4) the TF whose motif is disrupted has an open promoter in the same brain region(s) as the variant [168–170].

## Results

### Sample characteristics

**Ancestry.** In the SK data set, all samples with SNP data (all 251 children, 35% of the SK sample) had estimated proportion of EUR ancestry greater than 95%. Therefore, the SK data set (including parents) was assumed to be 100% European in genetic background. For the 532 people with SNP data in the UW data set (40% of the UW sample), 504,

15 and 2 individuals had EUR, AFR and East Asian (EAS) ancestry proportion greater than 95%, respectively. Eleven people were admixed (8 EUR/EAS and 3 EUR/AFR) and were counted as non-Europeans. Considering both self-reported ethnicity and SNP-estimated ancestry, 1208 people in the UW data set were assigned to the European category and 100 people to the non-European category. Self-reported ethnicity disagreed with SNP-estimated ancestry in only six of 783 samples where both were available (< 1%), suggesting self-report is reliable in these data sets. Twenty-five people categorized as unknown because data were unavailable were dropped from the analysis. The UH cohort had 5 African American individuals and 32 white Hispanic individuals as determined by self-report. The small number of individuals from non-European continental populations precluded meaningful analysis with a more finely stratified non-European ancestry variable. Exploratory analyses using only European samples resulted in findings similar to those presented here (data not shown).

**Traits.** In the tables and text that follow, these abbreviations are used for the tests and the processes they assess: **WID** (*WRMT-R Word Identification* for accuracy of oral reading of real words), **WA** (*WRMT-R Word Attack* for accuracy of oral reading of nonwords), **SP** (*WRAT-3* or *WRAT-R Spelling* for written spelling of orally dictated words), **SWE** (*TOWRE* speed of oral reading of real words), **PDE** (*TOWRE* speed of oral reading of nonwords), **NWR** (*CTOPP* Nonword Repetition for phonological memory). Table 1 shows sample sizes for UNADJ and VIQADJ traits for each dataset and the combined dataset. The probands (one per pedigree, by definition) were all children, but most of the remaining children were siblings of probands, with a few cousins in the UW sample. Probands account for ~29% of the largest analysis samples, and ~35% of the rest of the analysis samples. Phenotype and genomic data were collected on both the children and parents (except when noted). The variability in sample numbers included in the analyses reflects differences in the phenotyping protocols at the three institutions. VIQ scores were obtained for parents and children in the UW cohort, only for children in the SK cohort, and were not obtained for the UH cohort; therefore, the VIQADJ samples include only the UW and SK cohorts. For SWE and PDE, the UNADJ dataset is substantially larger than the VIQADJ dataset because VIQ was not available for SK parents. Results presented here in the main text focus on the larger, UNADJ, dataset except when findings are substantially different for VIQADJ.

S2 and S3 Tables contain demographic data for the samples used in the UNADJ and VIQADJ analyses with means and standard deviations for the traits in each group. The average VIQ score in the UW data set is almost a standard deviation higher than in the SK data set, as might be expected from the difference in sample selection between the two samples. This is supported by noting that the average score in the UW data set of children (109.7) is not significantly greater than that expected (106.4) from restricting enrollment to VIQ > 90 in a random sample. All the phenotypes have means

**Table 1. Sample size by data set for phenotypes analyzed.**

| Site | Pedigrees & size ranges[2] | Trait[1] | | | | | | | | | | | |
|---|---|---|---|---|---|---|---|---|---|---|---|---|---|
| | | UNADJ | | | | | | VIQADJ | | | | | |
| | | WID | WA | SP | SWE | PDE | NWR | WID | WA | SP | SWE | PDE | NWR |
| UW | 307 (1-19) | 1315* | 1315* | 1313* | 1305* | 1304* | 1302* | 1315* | 1315* | 1313* | 1305* | 1304* | 1302* |
| SK | 245 (1-5) | 335 | 336 | 338 | 713* | 711* | 337 | 331 | 332 | 338 | 332 | 332 | 333 |
| UH | 34 (1-6) | 34 | 34 | 0 | 34 | 34 | 0 | 0 | 0 | 0 | 0 | 0 | 0 |
| | **Total Subjects** | **1684** | **1685** | **1651** | **2052** | **2049** | **1639** | **1646** | **1647** | **1651** | **1637** | **1636** | **1635** |

[1]Word identification (WID) and word attack (WA) subsets of the Woodcock Reading Mastery Test, WRMT [126]; spelling (SP) subtest of the Wide Range Achievement Test – Revised, WRAT3-R [127]; single word reading efficiency (SWE) and phonological decoding efficiency (PDE) subtests of the Test of Word Reading Efficiency, TOWRE [128]; and non-word repetition (NWR) subtest of the Comprehensive Test of Phonological Processing [155]. Minor differences in samples size reflect occasional missing values.

[2]Number of pedigrees and pedigree size ranges (in parentheses) indicate pedigrees and ranges of number of individuals with both phenotype and genotype data analyzed in the current study

*Includes scores for parents

around zero because they are the residuals from a linear model. The means are not exactly zero because the adjustments were done on a larger data set than only the genotyped participants. Consideration of the SD column demonstrates that the traits fall into two categories: WID, WA and SP where the pre-adjustment value was a standard score, and SWE, PDE and NWR where the pre-adjustment value was a z-score. This difference is reflected in the magnitude of the effect sizes estimated for phenotypes in each category. Summary statistics for the residuals of the age-normalized phenotype measures used for non-VIQ adjusted analyses in all three samples are given in S4 Table.

### Association and bioinformatic analyses

Table 2 shows all common (MAF ≥ 0.01) variants that reached our stringent significance level with any trait. S5 and S6 Tables contain the *p*-values for aggregate testing of variants in and near each of the 5 genes with UNADJ and VIQADJ phenotypes respectively. Detailed results for single-marker testing of all SNPs with MAF > 0.01 are summarized in S7 Table (UNADJ phenotypes) and S8 Table (VIQADJ phenotypes).

**DYX1 on chromosome 15.** Of the two genes investigated in DYX1, only *CYP19A1* shows evidence of contribution of a common variant to any of the traits analyzed (Table 2). One variant downstream of *CYP19A1* was significantly associated with WID, WA and SP for both VIQADJ and UNADJ traits. The rarer allele was associated with an increase in performance on all measures. Aggregate testing of rare variants in *CYP19A1* and *DNAAF4* grouped by region (S5 and S6 Tables) did not reveal significant associations (*p* < 0.0025) with any trait. These analyses failed to implicate any exonic variants in either gene, common or rare, that were significantly associated with any trait (S7 and S8 Tables).

**DYX2 on chromosome 6.** A 211kb haplotype stretching from just downstream of *KIAA0319* to the second intron of *DCDC2* is associated with reduced performance on SWE:VIQADJ. Table 2 shows four variants (rs77743903,

**Table 2. Significant (*p* < 4.63 × 10⁻⁴) common (MAF ≥ 0.01) variants in *CYP19A1*, *DCDC2*, and *KIAA0319*.**

| TF Motif(s)[4] | rsID | Position[2] | REF/ALT (Freq) | Region[3] | Model | Trait[1] | | | |
|---|---|---|---|---|---|---|---|---|---|
| | | | | | | WID | WA | SP | SWE |
| | *CYP19A1* | | | | | | | | |
| MECOM | rs55712458 | 15:51,483,996 | G/C (0.198) | DS | UNADJ | 2.82 ($2.7 \times 10^{-6}$) | 2.45 ($1.0 \times 10^{-5}$) | 2.43 ($2.0 \times 10^{-5}$) | – |
| MECOM | rs55712458 | 15:51,483,996 | G/C (0.198) | DS | VIQADJ | 1.99 ($1.3 \times 10^{-4}$) | 1.95 ($1.7 \times 10^{-4}$) | 2.05 ($1.0 \times 10^{-4}$) | – |
| | *DCDC2* | | | | | | | | |
| – | rs77743903 | 6:24,332,778 | A/G (0.021) | I | VIQADJ | – | – | – | -0.46 ($4.6 \times 10^{-4}$) |
| LIN54, POU3F3, PHOX2B, POU2F3 | rs142310124 | 6:24,421,582 | A/C (0.023) | US | VIQADJ | – | – | – | -0.44 ($3.7 \times 10^{-4}$) |
| Nr2f6 | rs116652616 | 6:24,421,659 | G/A (0.023) | US | VIQADJ | – | – | – | -0.44 ($3.7 \times 10^{-4}$) |
| | *KIAA0319* | | | | | | | | |
| ELK1:HOXA1, Nrf1 | rs114979321 | 6:24,544,140 | A/G (0.031) | DS | VIQADJ | – | – | – | -0.40 ($1.5 \times 10^{-4}$) |

Transcription factors are listed with the SNP that likely disrupts its binding. Effect size (*p*-value) for dose of the rarer allele, - indicates non-significance. Shaded cells denote a haplotype.

[1]Traits as described in Table 1.

[2]Build GRCh37/hg19

[3]DS: downstream, I: intronic, E: exonic, US: upstream

[4]From JASPAR [166]

rs142310124, rs116652616, and rs114979321) that are significantly associated with reduced performance on SWE:VIQADJ. There is also suggestive evidence of association of these variants with reduced performance on SWE:UNADJ ($p = 0.001$, $p = 0.0098$, $p = 0.0098$, and $p = 0.0007$, respectively, S7 Table). The two upstream-of-*DCDC2* variants in the middle of the region (rs142310124 and rs116652616) are in complete disequilibrium in 1KGP-EUR and 1KGP-AFR [171], with the rare alleles on the same haplotype. The intronic and downstream variants on either side are in strong LD with this pair (D´ = 0.940 and D´ = 0.939 respectively), with the rare alleles appearing almost exclusively with the rare alleles of the middle pair. Table 3 shows the results of individual haplotype dosage models. Haplotype 2, which carries all four rare alleles and has a frequency of 1.5% in 1KGP-EUR, is associated with reduced performance on SWE:VIQADJ ($p = 2.1 \times 10^{-4}$). We cannot statistically distinguish the effects of individual variants because of the strong LD across the region.

Bioinformatic annotation indicates that rs142310124 is the best candidate as a causal variant on the haplotype. This variant is in a chromatin accessible region that is specific for neuronal cells in the nucleus accumbens and putamen and is predicted to disrupt motifs for four different TFs (POU2F3, POU3F3, PHOX2B and LIN54). LIN54 has an active/poised promoter in putamen, suggesting that rs142310124 affects reading performance by disrupting the binding of LIN54 in this tissue. The other three variants on the haplotype are not predicted to disrupt any TF motifs.

Aggregate testing of rare variants in DYX2 did not identify any significant ($p < 0.0025$) results in either *DCDC2* or *KIAA0319*. There is only suggestive evidence for an effect of intronic variants in *DCDC2* on SP:VIQADJ ($p = 0.0026$, S6 Table) and of 5´ variants in *KIAA0319* on SWE:VIQADJ and PDE:VIQADJ ($p = 0.0031$ and $p = 0.0033$ respectively, S6 Table).

**GRIN2B on chromosome 12.** No common SNPs in or near *GRIN2B* reach significance with any UNADJ or VIQADJ traits, but aggregate testing indicates an association of rare exonic variants (listed in Table S9) with both SP:UNADJ ($p = 0.00247$, S5 Table) and SP:VIQADJ ($p = 0.00058$, S6 Table). There were 11 missense variants, all of which were predicted by SIFT to be tolerated. One missense variant that was probably damaging according to PolyPhen was only present in 3 copies, precluding further statistical analysis. Of the 40 exonic variants with MAF < 0.01, 29 are in the last exon – 18 in the coding region (all synonymous or tolerated missense variants) and 11 in the 3´ UTR. There is suggestive evidence that rare variation in the last exon alone (29 variants) is associated with both SP:UNADJ ($p = 0.0070$) and SP:VIQADJ ($p = 0.0061$). Aggregate testing of rare variants in the downstream region also gives suggestive evidence of association for the same phenotypes (SP:UNADJ, $p = 0.0063$, S5 Table and SP:VIQADJ, $p = 0.0055$, S6 Table).

**Table 3. Haplotype (rs77743904--rs142310124--rs116652616--rs114979321) models in *DCDC2-KIAA0319* associated with SWE:VIQADJ.**

| Individual haplotype dosage models | | | | | |
|---|---|---|---|---|---|
| Haplotype Model | SNP alleles | EUR freq. | Obs. freq (n = count) | Effect Estimate | p-value |
| Hap 1 | A-AG-A | 0.961 | 0.963 | 0.388 | $7.0 \times 10^{-5}$ |
| Hap 2 | **G-CA**-G | 0.015 | 0.016 (n = 53) | -0.526 | $2.1 \times 10^{-4}$ |
| Hap 3 | A-AG-**G** | 0.013 | 0.011 (n = 37) | -0.270 | 0.11 |
| Hap 4 | **G**-AG-A | 0.009 | 0.002 (n = 8) | -0.305 | 0.42 |
| Hap 5 | A-**CA**-G | 0.001 | 0.004 (n = 12) | -0.107 | 0.73 |
| Hap 6 | **G-CA**-A | 0.001 | 0.002 (n = 6) | 0.213 | 0.68 |

Effect estimates are from six individual haplotype dosage models. The rarer allele at each SNP is in **bold**.

## Discussion

Here we provide results of a comprehensive investigation of underlying genomic variation in and surrounding five genes with prior evidence for an inherited effect on endophenotypes of dyslexia risk. The MIP sequencing approach that we used allowed inclusion of many more participants and variants than have been previously considered in sequencing studies of dyslexia and provided an agnostic approach for identifying underlying causal variants. Reliance of previous studies on detectable linkage disequilibrium between a causal variant and a small number of nearby genotyped polymorphisms is a possible cause of conflicting results across laboratories [76,83,84]. In contrast, in the current study we evaluated much of the gene neighborhoods, focusing on sequence that had the greatest potential for bioinformatic interpretation: protein coding regions and potential regulatory sites upstream and downstream of the candidate genes. Variants evaluated span a wide allele frequency range and fall in both coding and non-coding DNA, and results obtained unify some previously discrepant results.

The variants that met thresholds for association and further bioinformatic consideration represent non-coding DNA, with no clearly pathogenic coding variants. Although limited to a small number of selected genes and gene neighborhoods, the results provide an initial prediction of the types of genomic variation that are likely to be more broadly identified through evaluation of DNA sequence in genome-scale studies of specific learning impairments such as dyslexia. We speculate that variation in coding sequence that results in dramatic alteration of protein structure or function, as is typical of Mendelian disorders, is unlikely to play a role in dyslexia. Such protein-coding variants generally are rare, with large impacts on the phenotype, and are subject to negative selection. Dyslexia is a phenotype that is only recognized in the presence of widespread need for literacy, and non-coding variants with subtle effects on gene expression or control are more likely to be relevant. Selection against such variants, with their weak effects on the phenotype, would have been relatively ineffective in the small populations that were typical until very recently in human history. Instead, stochastic effects, such as genetic drift, would have had a role in driving changes in allele frequencies.

Targeted sequencing of two genes in the dyslexia-risk locus *DYX1* on chromosome 15 provides no support for a role for *DNAAF4* in modulating performance on any tested trait but does implicate a common variant downstream of *CYP19A1*. This downstream variant (rs55712458) provides the most significant support for association of any variant in our study but does not overlap any TF motifs that are currently annotated. Available annotation of TFs is incomplete and continues to evolve. Future results may yet suggest a functional role for this variant. The variant we identified does not appear on any of the Illumina or Affymetrix chips [167], so it is not surprising that it has not been seen in previous GWAS results. *CYP19A1* was previously nominated as a dyslexia-risk gene through identification of the breakpoint of a t(2;15)(p12;q21) translocation that disrupted the promoter region of the gene in a person with dyslexia [37]. *CYP19A1* encodes an enzyme that converts C19 androgens to C18 estrogens and is responsible for local synthesis of estrogens outside of the reproductive system. In the brain it is expressed from prenatal stages to adulthood [172] in multiple cell types where it regulates synaptic plasticity and plays a role in cognition, memory and language, and many other functions [173,174]. A possible role for this gene in dyslexia and quantitative reading/spelling performance traits might involve sex hormones in the brain during development given the male to female skewing in affected status in dyslexia.

Targeted sequencing of two genes in the dyslexia-risk locus *DYX2* on chromosome 6 implicate a haplotype that stretches 211kb from the downstream region of *KIAA0319* to the second intron of *DCDC2* and is associated with reduced performance on timed real-word reading adjusted for VIQ. The haplotype lies between variants previously implicated in *DCDC2* [138] and *KIAA0319* [51] and is within 7kb of READ1, a highly polymorphic human specific variant that contains a variable number of ETV6 binding sites [113]. Identification of this family of haplotypes provides a potential unifying explanation for the previously discrepant association results obtained for variants in each of the two genes. This family of haplotypes provides a parsimonious explanation that involves a single segregating locus, although one that consists of more than one polymorphic nucleotide. The best candidate on the haplotype for a causal variant is rs142310124, which is predicted to interfere with the binding of LIN54 to the haplotype in neuronal cells of putamen.

LIN54 is a member of the evolutionarily conserved MuvB core complex. When bound by additional factors it will either form the DREAM or MMB complexes that control cell-cycle dependent gene repression or activation, respectively, by binding directly to gene promoters [175,176]. The variant rs142310124 alters the LIN54 motif (5′-TTYRAA- 3′) by a nucleotide substitution of the fifth residue that presumably would affect either DREAM or MMB complex binding. rs142310124 is located 63 kb upstream of the DCDC2 transcriptional start site suggesting a long-range regulation of gene expression, currently an unknown function of the DREAM/MMB complexes. Yet, recent ChIP-seq experiments targeting LIN54 in cultured cells revealed wide-spread complex binding beyond the immediate gene promoter raising the possibility of such activity [177].

Comparison of our haplotype results in the DYX2 region to those previously reported in *DCDC2* and *KIAA0319* was hampered by several factors. First, only three of the nine SNPs that identify those haplotypes were targeted in our study. This is because previous studies used tagging SNPs to investigate common variation, whereas we specifically targeted variation in open chromatin, reasoning that these regions are more likely to be functional. Second, the probes we included in the READ1 region performed poorly, likely related to the repetitive nature of the locus (S1 Fig); therefore, we were not able to investigate READ1 directly. Thus, it remains unclear whether the haplotype we identified represents a new finding that suggests a role for LIN54 in transcriptional regulation of genes in DYX2, or whether its apparent influence on timed real word reading is due to its proximity to READ1.

We found evidence that rare exonic variants in *GRIN2B* are associated with performance on a test of spelling from dictation alone. Nearly three quarters of the observed exonic variants are in the last exon, which includes both coding sequence and 3′-UTR. There is also suggestive evidence that rare variants downstream of *GRIN2B* may be associated with spelling ability. Our classification of variants as downstream was based on the primary transcript noted in Ensembl. A study using mouse RNA and northern blot analysis described extensive lengthening of the 3′-UTR in *Grin2b*, specifically in the brain [178]. They observed extension of the 3′-UTR to include a long intergenic non-coding RNA 14.9kb downstream. Thus, it is possible that some of the rare variants we annotated as downstream of *GRIN2B* are in fact in the 3′-UTR. The 3′-UTR is known to influence post-transcriptional regulation in neurons by affecting mRNA stability, subcellular localization and translation control [179]. *GRIN2B* was selected for the current study because it lies in a region with evidence of linkage for a phonological non-word memory trait in the UW cohort [61] and was associated with verbal memory phenotypes in two European dyslexia datasets [63,64]. While the UW cohort here includes the samples that gave a signal on chromosome 12p for non-word memory, the statistical analyses of the two data sets are different. The original finding was based on linkage analysis, which is sensitive to rare alleles. In the analysis presented here, rare variants are analyzed in aggregate, and the precise choice of grouping for variants can affect the results. In addition, this analysis includes two other cohorts, which may weaken the association that led to the initial finding. Nevertheless, spelling from dictation can be understood as an ability that relies heavily on memory, so our finding in this data set is appealing. *GRIN2B* encodes GluN2B, one of the glutamate-binding subunits of the tetrameric N-methyl-D-aspartate ionotropic glutamate receptors (NMDARs) that are important for neuronal development and plasticity [180]. GluN2B is highly expressed prenatally in the brain where it is involved in learning and working memory via its role in synaptic plasticity and enhanced long-term potentiation [181,182]. Pathogenic coding variants and deletions in *GRIN2B* also cause a spectrum of neurodevelopmental disorders [183–185]. Non-coding variants in *GRIN2B* have also been associated with short term and working memory, intelligence quotient and cognitive impairments in dyslexia [63,64] and with other cognitive and behavioral traits [186–188].

There are, of course, some limitations to our study. Although we were able to generate more sequence data on a larger sample than has previously been evaluated for dyslexia, the tradeoff was its limitation to a subset of short DNA segments within a small number of previously implicated regions. We therefore cannot comment on genes and genomic regions that fall outside of the regions investigated or sequence alterations such as structural rearrangements or copy number variants (*e.g.*, the READ1 polymorphism [137–139]), that would likely be missed by this short-read technique. Failure to detect

some variants of interest in any of the regions analyzed could also be explained by the limited sample size for carrying out association analyses. Our data in the regions investigated allowed evaluation of most DNA positions in the regions for which current understanding of molecular mechanisms allows bioinformatics interpretation about effects on initiation of transcription. Even so, current knowledge about normal human variation in the regulome is still incomplete, and we acknowledge that transcription factor families share binding motifs, making the definitive identification of specific transcription factors difficult. It is also possible that variants in other regulatory motifs, which can be quite distant from the coding portions of the genes, may hold the causative DNA alterations.

In summary, we provide evidence that variants in or near *DCDC2*, *KIAA0319*, *CYP19A1*, influence reading-related traits and *GRIN2B* influences spelling ability. This study, with the largest clinically evaluated dyslexia-related sample size to date, is the first to comprehensively investigate both coding regions and cis-acting regulatory regions of dyslexia candidate genes. This provides both statistical power and depth of sequence evaluation. These results argue strongly against the causative involvement of large-effect coding variants in any of the studied genes and instead support a potential role in transcriptional regulation that may alter the quantity of RNA produced or its location. These results also illustrate some of the challenges that the field will face in identifying causal variants that may act through gene regulation rather than alteration of protein sequences. Use of whole-genome sequence (WGS), especially long-read, would capture regulatory elements with fewer complications, including detection of alterations in repeat sequences that might reside in deep intronic or intergenic regions. However, the WGS approach adds significant cost that could critically limit the number of samples used. The most feasible approach to corroboration of variants and haplotypes of interest discussed herein will therefore require evaluation in other dyslexia sample sets followed by functional studies in an appropriate cell model to begin to determine biological relevance [189], an endeavor that is well beyond the scope of the current analysis.

## Supporting information

**S1 File. Supplemental Methods, S2 – S6 Tables, and S1 Figure.**
(DOCX)

**S1 Table. Targets and MIPs.**
(XLSX)

**S7 Table. Common variant tests UNADJ.**
(XLSX)

**S8 Table. Common variant tests VIQADJ.**
(XLSX)

**S9 Table. GRIN2B rare variants aggregate analysis of SP:UNADJ.**
(XLSX)

## Acknowledgments

We are grateful to the family members who volunteered their time to participate in the research. John Wolff, Hiep Nguyen, and Edith PA Fuerte provided excellent technical, computational, and bioinformatics assistance. We thank the many graduate student assistants who administered the test batteries.

## Author contributions

**Conceptualization:** Nicola H. Chapman, Patrick A. Navas, Michael O. Dorschner, Ellen M. Wijsman, Wendy H. Raskind.
**Data curation:** Nicola H. Chapman, Patrick A. Navas, Ellen M. Wijsman.

**Formal analysis:** Nicola H. Chapman, Michele Mehaffey, Ellen M. Wijsman.

**Funding acquisition:** Elena L. Grigorenko, Cathy L. Barr, Ellen M. Wijsman, Wendy H. Raskind.

**Investigation:** Patrick A. Navas, Karen G. Wigg, Kaitlyn M. Price, Oxana Y. Naumova, Elizabeth N. Kerr, Sharon L. Guger, Maureen W. Lovett, Elena L. Grigorenko, Cathy L. Barr, Ellen M. Wijsman.

**Methodology:** Nicola H. Chapman, Patrick A. Navas, Michael O. Dorschner, Ellen M. Wijsman, Wendy H. Raskind.

**Project administration:** Ellen M. Wijsman, Wendy H. Raskind.

**Resources:** Karen G. Wigg, Kaitlyn M. Price, Oxana Y. Naumova, Elizabeth N. Kerr, Sharon L. Guger, Maureen W. Lovett, Elena L. Grigorenko, Virginia Berninger, Cathy L. Barr, Wendy H. Raskind.

**Software:** Ellen M. Wijsman.

**Supervision:** Elena L. Grigorenko, Cathy L. Barr, Ellen M. Wijsman, Wendy H. Raskind.

**Writing – original draft:** Nicola H. Chapman, Patrick A. Navas, Michael O. Dorschner, Ellen M. Wijsman, Wendy H. Raskind.

**Writing – review & editing:** Nicola H. Chapman, Patrick A. Navas, Michael O. Dorschner, Elena L. Grigorenko, Virginia Berninger, Cathy L. Barr, Ellen M. Wijsman, Wendy H. Raskind.

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
