## [Decision Letter · Decision Letter 0]

24 Mar 2024

PONE-D-23-41888Targeted analysis of dyslexia-associated regions on chromosomes 6, 12 and 15 in large multigenerational cohortsPLOS ONE

Dear Dr. Raskind,

Thank you for submitting your manuscript to PLOS ONE. After careful consideration, we feel that it has merit but does not fully meet PLOS ONE’s publication criteria as it currently stands. Therefore, we invite you to submit a revised version of the manuscript that addresses the points raised during the review process.

Your manuscript has been reviewed by two experts in the field. They both stress the overall quality and importance of the research reported, and have some specific suggestions for further improvement. Hopefully, all these issues can be addressed in a revision and/or response letter.

Personally, I am not very familiar with the types of analyses reported. From my own reading, I feel like the manuscript is targeted to a very specific readership, with quite a lot of knowledge/experience about/with genetics research. You might want to consider small adjustments in order to make the paper more accessible to a broader audience, for example by stressing the added value of this approach in studying dyslexia, as well as some more information about the lines of evidence (e.g. familial aggregation, linkage and association studies, copy number scan, structural chromosome rearragements) and the effects of genes (e.g. ciliogenesis).

We look forward to receiving your revised manuscript.

Kind regards,

Madelon van den Boer

Academic Editor

PLOS ONE

Reviewers' comments:

Reviewer's Responses to Questions

**Comments to the Author**

1. Is the manuscript technically sound, and do the data support the conclusions?

Reviewer #1: Yes

Reviewer #2: Yes

2. Has the statistical analysis been performed appropriately and rigorously? 

Reviewer #1: No

Reviewer #2: Yes

3. Have the authors made all data underlying the findings in their manuscript fully available?

Reviewer #1: Yes

Reviewer #2: No

4. Is the manuscript presented in an intelligible fashion and written in standard English?

Reviewer #1: Yes

Reviewer #2: Yes

5. Review Comments to the Author

Reviewer #1: This is a well-written manuscript describing targeted analyses of previously reported dyslexia-associated genes and loci. Overall, the approach and methodologies are sound. The results, while not surprising, confirm the soundness of the approach, building on and expanding previous findings. The work is a significant contribution to the field which overall, lacks vigorous independent replication.

There are some concerns that should be addressed:

1. On page 13, lines 262 – 268, section on Ancestry adjustment. More details should be included in the main text, such as the percentage of participants with self-reported ancestry and available SNP array genotype data for KING estimation. Of note, only 251 children in the SK data set and 532 individuals in the UW data set have existing SNP array genotype data to estimate ancestry - the SNP-based ancestry of nearly half of the participants in the study is unavailable. The authors should make note of the potential inaccuracy of self-reported ancestry. In addition, a simple European/non-European indicator may not be adequate to adjust for potential population stratification. Sensitivity analyses that include a more specific ancestry assignment as a covariate are suggested.

2. On Page 13, lines 278-281, two different phenotype adjustments (age, sex, with/without VIQ) are described. On Page 13, lines 264-265, the authors also describe the use of a European/non-European indicator to adjust for ancestry. Please confirm the covariates adjusted in the analysis.

3. On page 14 lines 284-285, the authors describe using linear regression to adjust for covariates. Did the authors check whether the original phenotypes followed the assumptions of linear regression?

4. Was there an adjustment for study site in the analysis?

5. Was there an adjustment for SES?

6. On page 14 lines 289-290, they mentioned that they used “data set-specific variances” in their analysis. What is this exactly? Was it used to account for difference between datasets.

7. On page 15 lines 311-314, the authors mentioned that they used a p-value of 0.0025 which was derived by 0.01 divided by 20. But 0.01/20=0.0005. This is confusing. What exactly was used as the cutoff for aggregate tests?

8. The determination of threshold for the single-variant tests is unclear. The authors considered the number of candidates out of 25000 genes instead of the number of SNPs for testing. Since SNPs within the same gene are in high LD, the authors may need to consider the effective number of independent markers (Me) for the adjustment of multiple testing (Li, 2012).

9. The thresholds for both the single-variant test and the Burden test didn’t adjust for the number of phenotypes. Given the strong correlations between the phenotypes, perhaps the effective number of phenotypes should be considered.

10. The authors conducted rare variant analysis by grouping SNPs based on their location relative to each candidate gene. We also suggest grouping SNPs within each gene based on functional annotations, such as loss-of-function and deleterious missense variants.

Reference:

Li, M. X., Yeung, J. M., Cherny, S. S., & Sham, P. C. (2012). Evaluating the effective numbers of independent tests and significant p-value thresholds in commercial genotyping arrays and public imputation reference datasets. Human genetics, 131, 747-756.

Reviewer #2: # summary

In this study, Chapman et. al. examine three dyslexia regions of interest in chromosomes 15 (DYX1), 6 (DYX2), 12 through targeted sequencing in a large sample of over 2,100 individuals from multiple families.

The article is well written and accurate. There are a few minor details that should be checked, as speficied below.

# minor comments

- short title: edit it to make it self contained, for instance replace MIP with targeted

- abstract, line 60: why does the data support an oligogenic model (vs a polygenic model)?

- abstract, line 58: for consistency to how the rest of the results were reported, specify that the association between GRIN2B and spelling occurred with and without VIQ adjustement.

- introduction: page 5, line 93: when arguing for the locus in chromosome 12q, what does "most convincing" mean? this is better explained later on, both in the results and in the discussion, but I feel that it's necessary to mention the evidence supporting this locus as well when presenting the few selected genomic regions for this study.

- methods:

- it is mentioned that the participants were selected for studies of dyslexia, and that there were both probands and related individuals, but it is not clear how many dyslexic individuals were included in the study, or what the distribution of the quantitative phenotypes looks like.

- please specify how the genetic ancestry was defined for individuals with genotyping data was available. There is a reference in line 263 to "KING estimation", but it would be good to specifcy that this is referring to genetic ancestry estimation.

- I understand that some of the people that were sequenced did not have genotyping data (because genetic ancestry could not be defined for all the individuals), but

- page 8, lines 192-193: it is unclear whether parents were included in the present study (given the lack of phenotypic data on them?). If not, please state it clearly.

- methods/(supplementary information): please specify the number of variants called in total and within each gene.

- supplementary data 1: could you include the annotation of the selected smMIPs? i.e. the BOCA, and ENCODE characterization criteria that were used for selection.

- page 12, last paragraph. Please specify how many MIP sequence variants and samples passed QC.

- please also provide also references to speficic software and state the used versions.

- supplementary tables S6, S7: please also provide the allele count for the common variants.

- results:

- it would be interesting to include annotations from JASPAR into supplementary tables 6 and 7. For instance, the main text specified that rs142310124

is the best candidate for the DCDC2-KIAA0319 haplotype, because it's predicted to disrupt motifs for four different TFs. However, this data is not available to the reader (which may want to evaluate other potential annotations as well).

- Tables S6 and S7: please also specify what A1 and A2 are, and whether the frequency(A1) refers to the current sample of any reference population allele frequency.

- SNP rs55712458 is multiallelic (G/A/C). I think these other annotations should also be included somewhere.

- it would be good to mention prior targetted sequencing efforts in the introduction, and to discuss the lack of replication with the KIAA0319 SNP rs138160539

from Caly et al. (2023). At the moment this lack of replication is only mentioned in the results section (page 18, lines 386-387).

- page 22, last paragraph: the results referring to supplementary tables S4 and S5 refer to the SKAT-O analysis, but the methods sections mentioned both SKAT and SKAT-O (pages 14-15, lines 303-309). What are the results from the SKAT? if different to SKAT-O, does that inform about the potential assumptions (since the power of the tests is different, i.e. SKAT-O "is more powerful when most variants are causal and effects are in the same direction" (lines 307-308)?)

- discussion:

- I think it would be informative to related the current results to the recent larger-scale GWASes for dyslexia and reading-related quantitative traits (Doust et al., 2022 and Eising et al. 2022). What can we interpret from the current study and associations in common variants such as rs55712458 (MAF~0.2) that were not significant in those GWASes?

page 26, lines 513 onwards: the initial linkage report was with nonword repetition, while the current study finds an association with spelling but not nonword repetition (despite being partly the same sample?). Please provide a possible explanation for this.

6. PLOS authors have the option to publish the peer review history of their article (what does this mean? ). If published, this will include your full peer review and any attached files.

**Do you want your identity to be public for this peer review?** For information about this choice, including consent withdrawal, please see our Privacy Policy .

Reviewer #1: No

Reviewer #2: No

---

## [Author Response · Author response to Decision Letter 0]

6 May 2024

We thank the academic reviewer and other two reviewers for their thoughtful comments and suggestions. Below we provide details of the changes made to the manuscript in response.

The academic reviewer commented that the manuscript is “targeted to a very specific readership, with quite a lot of knowledge/experience about/with genetics research” and suggested we “consider small adjustments in order to make the paper more accessible to a broader audience, for example by stressing the added value of this approach in studying dyslexia, as well as some more information about the lines of evidence (e.g. familial aggregation, linkage and association studies, copy number scan, structural chromosome rearragements) and the effects of genes (e.g. ciliogenesis)”. PLoS One is unusual among journals for the enormous breath of its content. With respect to our manuscript, from our involvement in the community of scientists studying dyslexia and related disorders, we feel that people who do not have a deep knowledge of statistical and/or molecular genetics will still understand the overall approach, the advantage of more comprehensive sequencing, and the results. Adding even a brief description of each type of study - familial aggregation, linkage and association studies, copy number scan, and structural chromosome rearrangements – would lengthen the paper unnecessarily as we cite references to papers that use these approaches and describe them in detail. We do include a statement indicating the advantage/power of our sequencing approach over other types of investigations: “This technology allows us, in this current multi-site study, to investigate the potential role of variants of smaller effect size, non-coding variants, and sample heterogeneity as possible explanations for previous variable results in ROIs implicated in dyslexia.”

Reviewer #1: "This is a well-written manuscript describing targeted analyses of previously reported dyslexia-associated genes and loci. Overall, the approach and methodologies are sound. The results, while not surprising, confirm the soundness of the approach, building on and expanding previous findings. The work is a significant contribution to the field which overall, lacks vigorous independent replication."

We thank the reviewer for these comments, which clearly articulate our major goal in carrying out the project!

"There are some concerns that should be addressed:

1. On page 13, lines 262 – 268, section on Ancestry adjustment. More details should be included in the main text, such as the percentage of participants with self-reported ancestry and available SNP array genotype data for KING estimation. Of note, only 251 children in the SK data set and 532 individuals in the UW data set have existing SNP array genotype data to estimate ancestry - the SNP-based ancestry of nearly half of the participants in the study is unavailable. The authors should make note of the potential inaccuracy of self-reported ancestry. In addition, a simple European/non-European indicator may not be adequate to adjust for potential population stratification. Sensitivity analyses that include a more specific ancestry assignment as a covariate are suggested."

We have added the percentage of people with SNP-based ancestry in the results paragraph (p16). Self-reported ethnicity agrees well with SNP-based ancestry in this dataset, and we have added a line to that effect. We added two sentences on page 17 pointing out that the non-European populations are too small for a more stratified analysis and that sensitivity analyses done in the early stages of this work indicated that the simple Eur/non-Eur variable was appropriate.

"2. On Page 13, lines 278-281, two different phenotype adjustments (age, sex, with/without VIQ) are described. On Page 13, lines 264-265, the authors also describe the use of a European/non-European indicator to adjust for ancestry. Please confirm the covariates adjusted in the analysis."

Thank you for pointing this out, we have clarified that there was an ancestry adjustment in both models (page 13).

"3. On page 14 lines 284-285, the authors describe using linear regression to adjust for covariates. Did the authors check whether the original phenotypes followed the assumptions of linear regression?"

Previous analyses in the UW data set have demonstrated that these models are appropriate to these phenotypes. We have added a sentence and citations to this effect (page 14).

"4. Was there an adjustment for study site in the analysis?"

Yes. Adjustment for study site was described in the original manuscript in the first sentence in “association testing” page 14. Phenotypes were adjusted separately within each data set, which effectively allows for a site-specific effect. We have added a sentence to make this point explicit for non-statistical readers.

This effectively allows for site specific effects.

"5. Was there an adjustment for SES?"

No. This (and other potential environmental covariates) was not available. We added a statement to this effect on page 14.

"6. On page 14 lines 289-290, they mentioned that they used “data set-specific variances” in their analysis. What is this exactly? Was it used to account for difference between datasets."

The residual variance was allowed to differ between data sets. We have restated this (hopefully more clearly) on page 14.

"7. On page 15 lines 311-314, the authors mentioned that they used a p-value of 0.0025 which was derived by 0.01 divided by 20. But 0.01/20=0.0005. This is confusing. What exactly was used as the cutoff for aggregate tests?"

Thank you for pointing this out. This was a typographical error. Our target significance level was 0.05 and after Bonferroni correction for 20 tests, we have p<0.0025 as our threshold. We have corrected this on page 16 and in supplemental table 6.

"8. The determination of threshold for the single-variant tests is unclear. The authors considered the number of candidates out of 25000 genes instead of the number of SNPs for testing. Since SNPs within the same gene are in high LD, the authors may need to consider the effective number of independent markers (Me) for the adjustment of multiple testing (Li, 2012)."

From this comment, it appears that the reviewer may have been confused by our different corrections for multiple testing. We have two different corrections: one for the higher-frequency individual markers and one for the aggregate tests for the rare variants within the genes evaluated. To try to make this more understandable, we broke this topic out into its own paragraph at the end of the section and have attempted to clarify our multiple-test adjustments in both settings. In short: we do include LD block-effects for calibrating the significance of individual-variant tests through use of reference-sample thresholds, just as do GWAS in general. We do not include LD-block effects for the aggregate variant tests. Both situations take into account the number of genes/regions being independently tested.

For convenience we excerpt the edited text here:

We determined significance thresholds for statistical testing as follows. A significance threshold for single-variant tests must account for the effects of LD blocks (e.g., Li et al 2012). Such thresholds do not change with increasing marker density (van den Berg 2019), but do depend on the population involved, due to differences in LD between populations (Li et al 2012). A study using 1KGP Phase 3 data showed that in European samples, a genomewide threshold of 9.26 × 10-8 is most appropriate for single-variant tests with a target type I error rate of 0.05. We used this genomewide threshold and scaled it to account for the approximate fraction of the genome under analysis. The present study involved 5 genes, compared to approximately 25000 in a full GWAS, so we use p < 4.63 × 10-4 (25000/5 × 9.26 × 10-8) as a stringent significance cutoff for single-variant tests. This allows for both the number of genes evaluated, and the presence of LD blocks in those gene regions. For aggregate testing, since the gene-region is the unit of analysis independent of presence/absence of LD blocks in the gene-region, further adjustment for the number of LD-blocks is not warranted. To achieve a type I error rate of 0.05, we therefore used a p-value of 0.0025 as the cutoff for aggregate tests, motivated by dividing 0.05 by 20 for a simple Bonferroni correction using the number of gene-region tests performed (4 tests for each of 5 genes). We did not adjust test thresholds for analysis of multiple traits since current studies do not typically do so. The limited literature to date shows no evidence for an increased false-positive rate, with the advantages of using multivariate and/or pleiotropic models falling primarily on the side of potential increase of power to detect true, but weak, associations (Julienne et al 2021).

"9. The thresholds for both the single-variant test and the Burden test didn’t adjust for the number of phenotypes. Given the strong correlations between the phenotypes, perhaps the effective number of phenotypes should be considered."

We did not adjust test thresholds for analysis of multiple correlated traits since current studies do not typically do so. The limited literature to date shows no evidence for an increased false-positive rate, with the advantages of using multivariate and/or pleiotropic models falling primarily on the side of potential increase of power to detect true, but weak, associations (Julienne et al 2021).

The last two sentences of the edited paragraph we excerpt above (page 16) include this information.

"10. The authors conducted rare variant analysis by grouping SNPs based on their location relative to each candidate gene. We also suggest grouping SNPs within each gene based on functional annotations, such as loss-of-function and deleterious missense variants."

The following comment is for the reviewer; no edits were made to the text:

The vast majority of the variants we identified are in non-coding sequence (by design, we sequenced exons and regions of open chromatin). Those in non-coding sequence are not simple to annotate – combing available databases is time consuming and typically involves investigator examination of one SNP at a time. Thus, it is not easy to sort non-coding SNPs by annotation. Also, there were very few exonic variants in any of the genes considered, so binning based on SIFT or polyphen predictions would have resulted in sample sizes that are too small to be informative.

"Reviewer #2: # summary

In this study, Chapman et. al. examine three dyslexia regions of interest in chromosomes 15 (DYX1), 6 (DYX2), 12 through targeted sequencing in a large sample of over 2,100 individuals from multiple families.

The article is well written and accurate. There are a few minor details that should be checked, as specified below.

# minor comments

- short title: edit it to make it self contained, for instance replace MIP with targeted"

Done, thank you.

"- abstract, line 60: why does the data support an oligogenic model (vs a polygenic model)?"

There is no sharp boundary between these two models. However, in the discipline of quantitative genetics that coined these terms, a polygenic model implies that genetic variation attributable to variation in individual genes (or other inherited units) is miniscule, and most genes contribute a tiny bit to the trait variance. In contrast, an oligogenic model implies multiple genes that may contribute through small, but non-trivial, contributions to the trait variance. Thus, what we find: - small number of sites that provide measurably different allelic effects – is more compatible with the oligogenic than the polygenic model.

"- abstract, line 58: for consistency to how the rest of the results were reported, specify that the association between GRIN2B and spelling occurred with and without VIQ adjustment. "

Thank you, we clarified this.

"- introduction: page 5, line 93: when arguing for the locus in chromosome 12q, what does "most convincing" mean? this is better explained later on, both in the results and in the discussion, but I feel that it's necessary to mention the evidence supporting this locus as well when presenting the few selected genomic regions for this study."

We edited this sentence for clarity.

A region on chromosome 12p provides evidence for linkage for a phonological non-word memory phenotype in the UW cohort (49), and harbors variants associated with dyslexia in other data sets (118-120).

"- methods:

- it is mentioned that the participants were selected for studies of dyslexia, and that there were both probands and related individuals, but it is not clear how many dyslexic individuals were included in the study, or what the distribution of the quantitative phenotypes looks like."

There is information in the supplement regarding proband qualification, which is as close as we can get to a binary dyslexia diagnosis. We have copied the information about proband definition into the main body of text, in both UW and SK cohorts. References are provided for previous analyses of both of these data sets. Information about proband definition was already there for the Houston cohort, and it is now highlighted.

"- please specify how the genetic ancestry was defined for individuals with genotyping data was available. There is a reference in line 263 to "KING estimation", but it would be good to specifcy that this is referring to genetic ancestry estimation."

We added the word “ancestry” to this line (now 285) to clarify.

"- I understand that some of the people that were sequenced did not have genotyping data (because genetic ancestry could not be defined for all the individuals), but

- page 8, lines 192-193: it is unclear whether parents were included in the present study (given the lack of phenotypic data on them?). If not, please state it clearly."

All individuals with both phenotype and genotype data were included in the analysis, as described in Table 1. To clarify we added a sentence to page 8.

Children and related adults with both trait and genotype data were included in the analyses.

"- methods/(supplementary information): please specify the number of variants called in total and within each gene."

We have added this information to the last paragraph before “Statistical and Bioinformatic Analyses”.

"- supplementary data 1: could you include the annotation of the selected smMIPs? i.e. the BOCA, and ENCODE characterization criteria that were used for selection."

Thank you for this suggestion – we have added a second tab to supplementary table S1 which lists the targeted genomic regions and their annotations as either exonic or identified by either BOCA or ENCODE.

- page 12, last paragraph. Please specify how many MIP sequence variants and samples passed QC.

We have added the number of variants passing QC in each of the 5 genes to the last paragraph before “Statistical and Bioinformatic Analyses”. The number of samples passing initial QC is already listed there, and the total number successfully genotyped for each variant is now listed in the Supplemental Tables S6 and S7.

"- please also provide also references to specific software and state the used versions"

Thank you for this reminder – we have added citations for the software used in the Statistical and Bioinformatics Analyses section.

"- supplementary tables S6, S7: please also provide the allele count for the common variants."

Thank you for pointing out this omission. We have added two columns giving the total number of samples both phenotyped and genotyped, and the minor allele count, for each of the six phenotypes. This adds 12 columns to the table, but as it is an excel file, readers can manipulate it as they wish.

"- results:

- it would be interesting to include annotations from JASPAR into supplementary tables 6 and 7. For instance, the main text specified that rs142310124 is the best candidate for the DCDC2-KIAA0319 haplotype, because it's predicted to disrupt motifs for four different TFs. However, this data is not available to the reader (which may want to evaluate other potential annotations as well)."

Thank you for catching this omi

---

## [Decision Letter · Decision Letter 1]

18 Jul 2024

PONE-D-23-41888R1Targeted analysis of dyslexia-associated regions on chromosomes 6, 12 and 15 in large multigenerational cohortsPLOS ONE

Dear Dr. Raskind,

Thank you for submitting your manuscript to PLOS ONE. After careful consideration, we feel that it has merit but does not fully meet PLOS ONE’s publication criteria as it currently stands. Therefore, we invite you to submit a revised version of the manuscript that addresses the points raised during the review process.

One of the previous reviewers is satisfied with the changes made to the manuscript. Unfortunately, the other previous reviewer was unavailable. Instead, a new reviewer has been able to read the manuscript. As I feel that the suggested changes, especially regarding the introduction of the study and discussion of the findings, can be accommodated and would further strengthen the manuscript, I encourage you to revise the manuscript according to the reviewer's suggestions. The reviewer also suggests additional analyses. If it would be possible to run these and include them in the manuscript as suggested, I believe that would be very helpful. However, I understand that the authors might feel otherwise. Alternatively, authors should clarify the criteria used to define dyslexia across cohorts and discuss the potential implications thereof.

We look forward to receiving your revised manuscript.

Kind regards,

Madelon van den Boer

Academic Editor

PLOS ONE

Journal Requirements:

Reviewers' comments:

Reviewer's Responses to Questions

**Comments to the Author**

1. If the authors have adequately addressed your comments raised in a previous round of review and you feel that this manuscript is now acceptable for publication, you may indicate that here to bypass the “Comments to the Author” section, enter your conflict of interest statement in the “Confidential to Editor” section, and submit your "Accept" recommendation.

Reviewer #2: All comments have been addressed

Reviewer #3: (No Response)

2. Is the manuscript technically sound, and do the data support the conclusions?

Reviewer #2: Yes

Reviewer #3: No

3. Has the statistical analysis been performed appropriately and rigorously? 

Reviewer #2: Yes

Reviewer #3: Yes

4. Have the authors made all data underlying the findings in their manuscript fully available?

Reviewer #2: No

Reviewer #3: Yes

5. Is the manuscript presented in an intelligible fashion and written in standard English?

Reviewer #2: Yes

Reviewer #3: Yes

6. Review Comments to the Author

Reviewer #2: The authors have clarified and addressed all my comments satisfactorily. I have no further comments.

Reviewer #3: Thank you for the opportunity to review the manuscript by Chapman et al. Please note that I was not involved in the original evaluation, and therefore, I am looking at this manuscript as a fresh submission.

My main concerns are around the study design and results interpretation. I provide some suggestions on how to address these issues.

1. The study conducts a deep sequencing analysis for five candidates genes selected because of previous "strong support" (line 89) from the literature. However, it's essential to acknowledge that previous associations at these loci may have derived from small and underpowered studies, which did not replicate in more recent GWAS studies. This is a commonly reported issue across various traits, including psychiatric conditions. E.g. see https://www.ncbi.nlm.nih.gov/pmc/articles/PMC8136395/

The authors recognise that there are inconsistent results in the literature but the manuscript should further emphasize the potential limitations of the original discovery studies and recognize the possibility that the associations in the selected genes could be false positives.

2. The explanation offered at lines 108-112 to explain the different outcomes across studies is not convincing. First, it needs to be supported by references and second, while different criteria could have affected results outcome, the small sample sizes remains the most plausible explanation.

3. Furthermore, the criteria for selecting these genes could be elucidated more clearly in the manuscript. While the study analyses 5 genes, the introduction describes previous associations only for DNAAF4, KIAA0319 and DCDC2. The rational for selecting GRIN2B and CYP19A1 is mentioned only later in the discussion. From what reported, the evidence for CYP19A1 is limited to one breakpoint study and GRIN2B was selected because of previous association in the UW cohort. As described by the authors, GRIN2B has been reported for associations with a range of cognitive traits including in GWAS, so it is possibly the gene out of the five selected is the one supported by the most strongest associations but these would not be specific to dyslexia.

4. My suggestion is to reframe the study and explain the rational for selecting all five genes in the introduction. It is essential to be completely transparent around the weak evidence supporting these genes, that nonetheless featured prominently in the field of dyslexia genetics. More explicitly, I would not start from the assumption that these genes are strong candidates but I would reframe the study as a comprehensive replication to further assess their potential role.

5. Another issue is around the criteria for defining dyslexia. While it appears that the FLDRC and SickKids cohorts used the similar criteria based on cut-off on IQ and reading measures, the UW cohort applied a different criteria that consider IQ discrepancies. I do not agree with this criteria but I do understand that this is a topic open for debate. My suggestion would be to state clearly how many participants had scored below -1.5 SD from the mean on reading measures as in the two other cohorts.

6. Ideally, the statistical analysis should be repeated with the exclusion of the participants that did not meet these criteria and presented in the supplementary material for the benefit of the reader that would have different views on the definition of dyslexia.

7. Finally, my main interpretation of the results is that the study does not robustly support the role of these genes in dyslexia, consistent with the interpretation that the initial discovery studies for these genes were false positives. While it is worth reporting the trends of observed associations, overall there is no compelling evidence. By addressing point 4) above and spelling out the weak associations in support of these genes, the main conclusion of the present study seems to be that these genes are unlikely to play a major role in dyslexia.

8. The latest GWAS for dyslexia have demonstrated the highly polygenic nature of this condition. Therefore, it would be unlikely to find a few genetic factors playing a major role in many individuals. Exactly the same scenario of high polygenicity is emerging from pretty much most human complex traits. Therefore, rather than interpreting the results on the assumption that the selected genes are expected to play a major role with a specific class of genetic variants, it is necessary to recognise the highly polygenic nature of dyslexia (as opposed to the proposed “oligenic” model – line 60) and to contextualize the findings within our current understanding of the field of dyslexia, and more generally complex traits, genetics.

7. PLOS authors have the option to publish the peer review history of their article (what does this mean? ). If published, this will include your full peer review and any attached files.

**Do you want your identity to be public for this peer review?** For information about this choice, including consent withdrawal, please see our Privacy Policy .

Reviewer #2: No

Reviewer #3: No

---

## [Author Response · Author response to Decision Letter 1]

22 Aug 2024

Comment from Dr. van den Boer

“The reviewer also suggests additional analyses. If it would be possible to run these and include them in the manuscript as suggested, I believe that would be very helpful. However, I understand that the authors might feel otherwise. Alternatively, authors should clarify the criteria used to define dyslexia across cohorts and discuss the potential implications thereof.”

We do not feel that additional analyses would overcome the limitations engendered by inclusion of cohorts from multiple study groups. Differences in ascertainment, inclusion criteria and phenotyping batteries are part and parcel of such collaborations. As such, covariate adjustments are a standard and expected approach to data analysis. We included such adjustments for cohort, with and without adjustment for VIQ, which should address differences in ascertainment by site as well as inclusion criteria for probands. We also did not carry out analyses of dyslexia per se, but instead analyzed the phenotypes as quantitative traits. Therefore, a description of details of dyslexia diagnosis across the cohorts does not have an obvious goal that is relevant to interpretation of the analyses. Details of ascertainment for the two larger studies are also available in cited papers, so any description in this paper would be duplicative. Analyzing only UW families in which the proband met a non-discrepancy criterion would also not be illuminating as the smaller sample size could simply lead to false negative results on the basis of sample size alone.

Comments from the new reviewer

1. “The study conducts a deep sequencing analysis for five candidate genes selected because of previous "strong support" (line 89) from the literature.”

We actually did not, and do not, state that the candidate loci had particularly strong support. We agree with the reviewer that this would be a misstatement of the evidence. The phrase “a small number have the strongest support’ was meant as a comparative. To address the reviewer's comment, since other readers may jump to the same conclusion, we changed that phrase to “a small number have received support by more than one group.” We also added a sentence in the last paragraph of the Introduction to clarify the choices of loci for analysis, “The analyses focused on two highly cited loci and a genomic region implicated by our previous studies and supported by the literature.”

“However, it's essential to acknowledge that previous associations at these loci may have derived from small and underpowered studies, which did not replicate in more recent GWAS studies.”

Failure to detect some variants of interest in any of the regions analyzed could also be explained by the limited sample size for carrying out association analyses.

“The authors recognise that there are inconsistent results in the literature but the manuscript should further emphasize the potential limitations of the original discovery studies and recognize the possibility that the associations in the selected genes could be false positives.”

We specifically cited failures to support the results of each type of study – linkage, association and GWAS. The concern about false positives holds for every linkage study, every GWAS, and every other statistical test, so it is not clear what the reviewer is asking for. Of these, when carried out properly, linkage analysis has a low false positive rate in sample sizes that are far less than those needed to carry out GWAS. But we undertook this comprehensive sequencing study precisely to study the question of “false positives”. As none of the previous sequencing studies of these genes/loci evaluated this large a sample nor included noncoding DNA the possibility of an undetected causative variant in any of the candidate genes could not be excluded. To be more explicit, we added a phrase and several sentences (highlighted below in yellow).

While support for involvement of the aforementioned genes has been reported from both a variety of association and linkage analyses and functional studies, evidence favoring particular genes in the ROIs is inconsistent or difficult to interpret [69-75]. There have been failures to detect linkage [76-78] or association [79-85], as well as reports of increased risk attributed to opposite alleles [50, 51, 79]. For a complex trait there is also the chance that composite quantitate trait loci (QTLs) are responsible for some of the linkage analysis results [86, 87]. False-positive results are another possible explanation. Demonstration of potential functional competence of the putative risk allele in an animal model is also difficult to interpret in the context of a human trait [88].

2. “The explanation offered at lines 108-112 to explain the different outcomes across studies is not convincing. First, it needs to be supported by references and second, while different criteria could have affected results outcome, the small sample sizes remains the most plausible explanation.”

We have reworded this portion of the paragraph and provided references.

“Variability in conclusions across the different study designs and samples is common and not surprising. Genetic heterogeneity has been responsible for discrepant results since the earliest days of genome scans, even for “simple” Mendelian traits [100]. Genome-wide linkage analyses and GWAS both allow location scans, but with different localization resolution and sensitivities to less vs. more-common trait-gene allele frequencies [101], and with power to detect genetic effects influenced by sample ascertainment procedures [102].”

Different statistical approaches require different sample sizes of participants, so many of the linkage studies of dyslexia or component phenotypes were equivalent or greater in power under some circumstances to some of the GWAS studies in this regard. The largest GWAS was flawed in using only self-report. To address the reviewer's concerns, we have included this statement: “However, the large sample size was only feasible though use of cases without a clinical diagnosis. This a situation that can lead to statistical heterogeneity in results, raising concerns about usefulness of such samples, as has been reported in application to another complex trait [98]. “

3. “Furthermore, the criteria for selecting these genes could be elucidated more clearly in the manuscript. While the study analyses 5 genes, the introduction describes previous associations only for DNAAF4, KIAA0319 and DCDC2. The rational for selecting GRIN2B and CYP19A1 is mentioned only later in the discussion. From what reported, the evidence for CYP19A1 is limited to one breakpoint study and GRIN2B was selected because of previous association in the UW cohort. As described by the authors, GRIN2B has been reported for associations with a range of cognitive traits including in GWAS, so it is possibly the gene out of the five selected is the one supported by the most strongest associations but these would not be specific to dyslexia. My suggestion is to reframe the study and explain the rational for selecting all five genes in the introduction. It is essential to be completely transparent around the weak evidence supporting these genes, that nonetheless featured prominently in the field of dyslexia genetics.”

In the Introduction, we added the rationale for selecting GRIN2B and CYP19A1 and cite the relevant references. “CYP19A1, another candidate gene in the DYX1 locus [37, 52] is of interest because of the almost universally observed ratio imbalance of males:females with dyslexia; CYP19A1 codes for aromatase, an enzyme that converts androgens to estrogens in the brain [53].”

The previous version had “From the other four ROIs identified by our SNP linkage analyses in the UW cohort, we chose to include the gene for ionotropic glutamate receptor subunit 2B (GRIN2B)”. We substituted: “Our linkage analyses for various quantitative measures of dyslexia using the University of Washington cohort identified additional candidate loci [63-65]. One of the strongest linkage signals was in a region on chromosome 12p [64], which contains GRIN2B, a gene that had support as a dyslexia candidate gene from studies in other data sets [66-68].”

4. “More explicitly, I would not start from the assumption that these genes are strong candidates but I would reframe the study as a comprehensive replication to further assess their potential role.”

We do not understand what is requested beyond what we had already written about our intent in the Introduction. We added the word “possible” in this sentence, “We carried out a comprehensive analysis of the coding region and some regulatory element motifs of five putative dyslexia risk genes to assess their possible role.” The word “potential” had already been used in a sentence in the same paragraph, “to investigate the potential role of variants.”

5. “Another issue is around the criteria for defining dyslexia. While it appears that the FLDRC and SickKids cohorts used the similar criteria based on cut-off on IQ and reading measures, the UW cohort applied a different criteria that consider IQ discrepancies. I do not agree with this criteria but I do understand that this is a topic open for debate. My suggestion would be to state clearly how many participants had scored below -1.5 SD from the mean on reading measures as in the two other cohorts.”

Please see our response to Dr. van den Boer.

In the Methods section we provided more explanation of how our strategy, including analyses of quantitative traits, minimized the problems inherent in joint analyses of independently collected study groups. “This strategy provided a large total number of participants screened with an essentially equivalent test-battery allowing for joint analysis across the three cohorts while minimizing introduction of excess phenotypic heterogeneity that may be introduced by mixing multiple phenotypic measures and including a threshold to create a binary affected/unaffected outcome. Even so, some variability in underlying risk allele frequencies is expected across cohorts here because details of recruitment invariably differ across recruitment sites, as is the case for virtually all analyses that aggregate data from multiple sites. This may lead to biased estimates of risk-allele effects (115) but does not affect interpretation of the results of hypothesis testing.”

6. “Ideally, the statistical analysis should be repeated with the exclusion of the participants that did not meet these criteria and presented in the supplementary material for the benefit of the reader that would have different views on the definition of dyslexia.”

Please see our comment to Dr. can den Boer.

7. “Finally, my main interpretation of the results is that the study does not robustly support the role of these genes in dyslexia, consistent with the interpretation that the initial discovery studies for these genes were false positives. While it is worth reporting the trends of observed associations, overall there is no compelling evidence. By addressing point 4) above and spelling out the weak associations in support of these genes, the main conclusion of the present study seems to be that these genes are unlikely to play a major role in dyslexia.”

We do not state that the genes described here either do or do not play a major role in dyslexia. Our study here is not appropriate for evaluating this question. We only report that there is strong statistical support for some role in the quantitative traits evaluated for some variants investigated here. Regardless, our study by itself does not negate the candidate locations as different genes might be responsible for the signals in these loci obtained in previous studies.

8. “The latest GWAS for dyslexia have demonstrated the highly polygenic nature of this condition. …. Therefore, rather than interpreting the results on the assumption that the selected genes are expected to play a major role with a specific class of genetic variants, it is necessary to recognise the highly polygenic nature of dyslexia (as opposed to the proposed “oligenic” model – line 60) and to contextualize the findings within our current understanding of the field of dyslexia, and more generally complex traits, genetics.”

Even if multiple genes participate in a phenotype, this does not preclude identification of one or more of them that contribute enough to be detected above the rest. GWAS and linkage analyses have different strengths and usages; the GWAS results do not negate the results of the linkage analyses and vice versa. Rather than using either oliogogenic or polygenic in the Abstract we have substituted “multigenic”.

Thank you both for your comments. We think our changes in response to them have improved the manuscript.

---

## [Decision Letter · Decision Letter 2]

12 Nov 2024

PONE-D-23-41888R2Targeted analysis of dyslexia-associated regions on chromosomes 6, 12 and 15 in large multigenerational cohortsPLOS ONE

Dear Dr. Raskind,

Thank you for submitting your manuscript to PLOS ONE. After careful consideration, we feel that it has merit but does not fully meet PLOS ONE’s publication criteria as it currently stands. Therefore, we invite you to submit a revised version of the manuscript that addresses the points raised during the review process.

To be honest, it is quite hard for me to reach a decision on this manuscript. Overall, you have adequately responded to the issues raised by me and the final reviewer and a previous reviewer was already in favor of acceptance of the manuscript. However, on some issues we, as well as you and the reviewer, continue to disagree. The two main issues now are:

More clarity is needed about the inclusion criteria across the different samples. One of the samples seems to have applied a discrepancy criterion, whereas the other two focused on severity of the literacy impairments. This should be explicitly mentioned. The authors argue that this difference is covered by adding VIQ to the analyses, but that is incorrect. It is not IQ measures that are likely to differ, but literacy scores. Through a discrepancy criterion persons with less severe reading and/or spelling problems might have been included in one sample as compared to the others. The implications of these differences should be discussed in the manuscript. In addition, I would appreciate more information in general on the ranges of scores obtained in the included participants. Authors argue that an important limitation of previous studies is that participants were included without a formal diagnosis of dyslexia. However, this also seems to be the case in the current study. As families were included, not every member of the family would have dyslexia or risk scores on literacy right? I realize that this last question is probably related to me being rather unfamiliar with this type or research. However, as this is probably the case for more readers, I would still appreciate some more information on this in the manuscript, without readers having to resort to previous publications.I think it is now sufficiently clear in the introduction that previous evidence on dyslexia-related genes is weak at best and that the current study is focused on some that have previously been associated with dyslexia. However, what is missing is a theoretical or more fundamental understanding of why particularly these ROIs would be of interest. This has been added for some of the ROIs, but not all. To me, it now comes across as a rather random selection of RIOs, whereas this is probably not the case. See also points 1 and 2 of the reviewer. Minor issue: how should the final sentences of the introduction be interpreted? Is this the hypothesis or a preview of the results?

I would like to invite the authors to address these two issues in a final minor revision of the manuscript.

We look forward to receiving your revised manuscript.

Kind regards,

Madelon van den Boer

Academic Editor

PLOS ONE

Journal Requirements:

Reviewers' comments:

Reviewer's Responses to Questions

**Comments to the Author**

1. If the authors have adequately addressed your comments raised in a previous round of review and you feel that this manuscript is now acceptable for publication, you may indicate that here to bypass the “Comments to the Author” section, enter your conflict of interest statement in the “Confidential to Editor” section, and submit your "Accept" recommendation.

Reviewer #3: (No Response)

2. Is the manuscript technically sound, and do the data support the conclusions?

Reviewer #3: No

3. Has the statistical analysis been performed appropriately and rigorously? 

Reviewer #3: Yes

4. Have the authors made all data underlying the findings in their manuscript fully available?

Reviewer #3: No

5. Is the manuscript presented in an intelligible fashion and written in standard English?

Reviewer #3: Yes

6. Review Comments to the Author

Reviewer #3: I thank the authors for addressing my comments, however the key assumption of the manuscript remains problematic.

Specifically:

1) The criteria for gene selection is still unclear and unconvincing. Although, I appreciate that the authors now specify at the end of the introduction that the genes were selected as “highly cited loci” the abstract also states that the genes were selected on the bases of prior evidence of “association from more than one samples”.

Such criteria and such evidence have not been fully clarified. Specifically, there seem to be confusion between broad loci identified through linkage analysis and genes proposed via candidate gene association studies.

For example, multiple studies reported linkage at DYX1, but only one study reported association for the CY19A1 gene.

2) My impression is that the five genes were mainly selected because reported in the literature, but no specific criteria were applied.

If that is the case, this needs to be spelt out more clearly.

3) My view is that the evidence supporting these genes is very weak and the revised manuscript and the response of the authors have not changed my position.

In particular, when highlighting that these genes are not supported by GWAS results, the authors suggest in their response that the selected genes have been identified via linkage studies for which large samples are not necessary. This is problematic because, while most of these genes were selected as candidates for being located within linked regions, they were tested mainly through association analysis. It is now well established that association studies in small samples are likely to lead to false positive results.

Furthermore, in their revision the author argues that the lack of clinical diagnosis could have affected the large GWAS by Doust et al. However, earlier GWAS that used both clinical diagnosis (Gialluisi et al 2021 https://www.nature.com/articles/s41380-020-00898-x) and quantitative measures (Gialluisi et al 2019) also failed to provide support for the genes selected here. Notably these GWAS analysed samples that led to the identification of some of the dyslexia linked regions.

Therefore, the revised interpretation for the lack in GWAS of support for the selected genes is not convincing and needs to be revisited.

4) In line with the weak evidence supporting these genes, my interpretation of the results is that, as expected, the present study does not support their role in dyslexia.

Other points:

- Please, avoid referring to dyslexia as a “disability” or “disorder” and use terms like “difficulty”.

- The submission states that “All data are fully available without restriction”, but I was not sure where to find them.

7. PLOS authors have the option to publish the peer review history of their article (what does this mean? ). If published, this will include your full peer review and any attached files.

**Do you want your identity to be public for this peer review?** For information about this choice, including consent withdrawal, please see our Privacy Policy .

Reviewer #3: No

---

## [Author Response · Author response to Decision Letter 2]

19 Mar 2025

ITEMIZED RESPONSES TO THE CRITIQUES

Comment from Dr. van den Boer

1. More clarity is needed about the inclusion criteria across the different samples. One of the samples seems to have applied a discrepancy criterion, whereas the other two focused on severity of the literacy impairments. This should be explicitly mentioned.

In the section in “METHODS” on sample acquisition we have from the beginning specified the different inclusion criteria for probands. For additional clarity, in the first paragraph of the section “Sample and phenotype selection strategy” we added citations that provide detailed ascertainment and inclusion strategies for the University of Washington (UW) and The Hospital for Sick Children cohorts. In the individual site paragraphs, we refer to the “Supplement” where details from those publications can be found and mention the tests that were used in the qualification process. For example, the statement for the UW site reads: “Recruitment and evaluation …..are comprehensively described elsewhere, are briefly summarized herein, and are provided in more detail in the Supporting Information document.”

After that statement, as requested, we added this sentence: “For the UW cohort a discrepancy criterion was used for qualification of a child as a proband.”

2. The authors argue that this difference is covered by adding VIQ to the analyses, but that is incorrect. It is not IQ measures that are likely to differ, but literacy scores. Through a discrepancy criterion persons with less severe reading and/or spelling problems might have been included in one sample as compared to the others. The implications of these differences should be discussed in the manuscript.

There seems to be a confusion about both the rationale and analysis steps that we used and the points we were trying to make. In no case did we analyze IQ measures against any of the genomic variants. In all our association testing we look at correlation between variation in the covariate-adjusted literacy scores and the DNA variants/genotypes. To clarify what we did, we have re-organized and re-edited a number of paragraphs in the section “Statistical and Bioinformatic Analyses”. For readers who are not familiar with the standard statistical methods used in genetic epidemiology we added an overview that contains a brief description.

The section “Phenotypes and adjustments” has been extensively edited and rearranged to make it easier for those without substantial statistical expertise to understand when each component of the modeling is introduced.

Regarding the concern about use of a discrepancy score in probands’ ascertainment in only one site, this is only one example of an ascertainment differences among the three sites, as is typical of virtually all multisite projects. For this reason, we do our covariate adjustment within site to account for any site-specific effect. After all, the age-normalized score is an example of this to begin with - the actual reading scores that we start with are already re-scaled to reflect age- or grade-based expectations. However, there is ample reason and evidence in the literature to expect that VIQ measures may be correlated with the literacy scores, justifying investigation or comparison of the effect of association of VIQ-adjusted vs. not-adjusted literacy scores with genomic variants in the region. The value or necessity of including (or not) IQ or VIQ has different "camps" in the dyslexia research community, and we are not married to either camp. We feel that especially as a PLoS One submission it is best to provide both analyses, although our preference is to lean more heavily on results from the UNADJ analyses with their larger sample sizes.

To the extent that VIQ and any of the traits are correlated, adding the VIQ of individuals to the analysis model does adjust for this sampling detail because in the end the analysis investigates association of the residuals from the linear mixed model (where the VIQ for an individual is a fixed effect covariate) and number of ALT alleles at each SNV. The VIQ is never evaluated directly in an association analysis to the SNV ALT allele count, as the critique seemed to imply.

A fact that is very important for interpretation of our results is that we did not assess association of the variants/genes to dyslexia but only to quantitative phenotypes (endophenotypes) used to assess reading ability/impairment. We think we have diminished the possible confusion by re-editing the analysis section and inserting clarifying phrases throughout. The first sentence of the “DISCUSSION” already read “Here we provide results of a comprehensive investigation of underlying genomic variation in and surrounding five genes with prior evidence for an inherited effect on endophenotypes of dyslexia risk.” We have added phrases in several places to make this more apparent throughout. The “ABSTRACT” now substitutes “traits” for the quantitative scores on measures used to assess dyslexia, “We did not analyze dyslexia per se”. In the “INTRODUCTION” we added a phrase to the sentence beginning “Here we carried out a comprehensive analysis … to assess their possible role”. It now ends with “in performance on six tasks commonly used in the evaluation for dyslexia.” In “METHODS”, under “Sample and phenotype selection strategy” we state, “We only used the quantitative phenotypes and not the dyslexia diagnostic status for our analyses; henceforth, we use “trait(s)” to refer to these quantitative measures.” To the first sentence under “Statistical and Bioinformatic Analysis” we added: “We carried out a comprehensive association analysis of performance on six tasks commonly used in the evaluation for dyslexia, but did not analyze dyslexia per se.” For the remainder of the manuscript and supplement, we substituted “trait” for “phenotype” in all relevant places to refer to adjusted quantitative phenotypes.

In the “DISCUSSION”, to the sentence beginning “Targeted sequencing of two genes in the dyslexia-risk locus DYX1 on chromosome 15 provides no support for a role for DNAAF4” we added the phrase “in modulating performance on any tested trait”. And n the last sentence of the paragraph on CYP19A1, after “A possible role for this gene in dyslexia” we inserted the phrase “and reading/spelling performance on traits related to dyslexia”.

In “METHODS” at the beginning of the section “Phenotypes and adjustments”, we added an explanation for why we included participant sets ascertained with different criteria. “As with all complex traits for which there is some heterogeneity across collection sites, the resulting potential heterogeneity adds a cost to the sample size required to detect association by reducing variant effect size in the full sample. However, the only way to achieve sufficiently large samples to detect association with complex traits is to include as many existing sample sets as possible that have that have assessed the same traits.”

3. In addition, I would appreciate more information in general on the ranges of scores obtained in the included participants. Authors argue that an important limitation of previous studies is that participants were included without a formal diagnosis of dyslexia. However, this also seems to be the case in the current study. As families were included, not every member of the family would have dyslexia or risk scores on literacy right? I realize that this last question is probably related to me being rather unfamiliar with this type or research. However, as this is probably the case for more readers, I would still appreciate some more information on this in the manuscript, without readers having to resort to previous publications.

You are absolutely correct that not all family members have dyslexia, but all family members who were used in the analysis do have phenotype data for at least some of the traits in “Table 1”, and these data are used. It seems as if this nuance might have been unclear, despite the numbers and symbols in “Table 1”. We have added an additional column to “Table 1” to summarize the families a bit more and inserted more explicit statements to the “METHODS” under the “Statistical and Bioinformatic Analysis” section. As requested, we also added a new summary table to the supplement (Table S4) with more summary statistics for each of the "raw" (age-normed) variables.

In terms of subject sampling, choice of immediate (first-degree) family members for measurement of phenotypes and from whom to obtain samples for DNA isolation was purely based on those subjects' availability. It is not at all unusual in genetic studies that complete sibships and both parents are sampled and measured when the trait under study is identified through children. But our intent was not to write a review paper about how to carry out a genetic study, which is what it would take to cover every detail of how one does such studies. There is plenty of existing literature (and books) to help the novice get started.

Regarding the range of scores in all three cohorts it is important to consider that ascertainment was done through a proband, but parents and siblings were included regardless of history of dyslexia/reading difficulty or performance on the tests. The probands constitute the minority of subjects in the analyses.

To make it clear from the outset that we evaluated association of the genes with phenotypes used to assess dyslexia, not dyslexia itself, beginning with the “ABSTRACT”, we use the term “traits” for the quantitate phenotypes throughout.

In addition, in the last paragraph of the “INTRODUCTION” we reworded the sentence that previously read: “Here we carried out a comprehensive analysis of the coding region and some regulatory element motifs of five putative dyslexia risk genes to assess their possible role.” It now reads: “We report results from a comprehensive analysis of the coding regions and some regulatory element motifs of five putative dyslexia risk genes to assess their possible role in performance on six tasks that yield quantitative scores and are commonly used in the evaluation for dyslexia.” This change should clarify that we assessed dyslexia-related phenotypes, not dyslexia itself.

4. I think it is now sufficiently clear in the introduction that previous evidence on dyslexia-related genes is weak at best and that the current study is focused on some that have previously been associated with dyslexia. However, what is missing is a theoretical or more fundamental understanding of why particularly these ROIs would be of interest. This has been added for some of the ROIs, but not all. To me, it now comes across as a rather random selection of RIOs, whereas this is probably not the case. See also points 1 and 2 of the reviewer.

Comments from Reviewer 3

1 (a). The criteria for gene selection is still unclear and unconvincing. Although, I appreciate that the authors now specify at the end of the introduction that the genes were selected as “highly cited loci” the abstract also states that the genes were selected on the bases of prior evidence of “association from more than one samples”.

In the sentence excerpted above, our use of the word “association” was confusing because it can be used to imply “involved in some way” but has a specific meaning as a statistical approach. All candidate genes included in our study were identified/nominated through further research in regions initially identified through linkage analyses. To avoid this imprecise wording, we replaced the “of association” with “for a role”

1 (b). Such criteria and such evidence have not been fully clarified. Specifically, there seem to be confusion between broad loci identified through linkage analysis and genes proposed via candidate gene association studies. For example, multiple studies reported linkage at DYX1, but only one study reported association for the CY19A1 gene.

The “INTRODUCTION” lists the types of analyses that led to identification of the loci (linkage analyses, genome-wide association studies, copy number scan, structural chromosome rearrangements and whole genome sequencing). We now provide more information regarding the choice of the genes to target within the loci. This is discussed in our response to point 2 below.

2) My impression is that the five genes were mainly selected because reported in the literature, but no specific criteria were applied.

If that is the case, this needs to be spelt out more clearly.

This was a first foray into examining a relatively complete set of variants at the sequence level data in a large sample of learning disabilities data sets. We chose a strategy that balanced cost against the impossible task of a complete genome wide scan, which would have been too much to try for a complex trait at this point with this sample size. We realize that the organization of the manuscript made it difficult to grasp the reason for selection of the 12p locus containing the GRIN2B gene. In the “INTRODUCTION” we provide some more information about the loci and in the “METHODS” section “Overview of Rationale and Data Used” we added two paragraphs to explain the logic behind the number and choice of candidate genes. In the first paragraph contains this wording:

“Practical issues of number of genes investigated were driven by cost, sample size, and challenges of interpreting genomic sequence data in non-coding sequence. To maximize sample size, we sequenced every individual in our combined dataset who had the relevant phenotypic data. This strategy gave us capacity to evaluate five genes and regulatory/splice regions around those genes in three genomic regions.”

The second through fourth paragraphs give details of the choice of regions and genes and has additional literature citations.

3 (a). My view is that the evidence supporting these genes is very weak and the revised manuscript and the response of the authors have not changed my position.

In particular, when highlighting that these genes are not supported by GWAS results, the authors suggest in their response that the selected genes have been identified via linkage studies for which large samples are not necessary. This is problematic because, while most of these genes were selected as candidates for being located within linked regions, they were tested mainly through association analysis. It is now well established that association studies in small samples are likely to lead to false positive results.

We make two comments here. First, virtually all gene identification involves association analysis at some stage. This is only one of the possible steps and processes used to get from a genome region to a gene. There are the rare exceptions where a mutation falls in an obvious gene, but because we don’t know a lot about what most genes do, this is an uncommon situation. Second, the comment about small samples and false positive results is not complete. Yes, one can get false positive results, but also one can get false negative results. One might simply not recognize the false negative results because of how one does literature searches, and false positive results are more likely to make it to publication. But the statement as it stands is incorrect at the core of statistical analysis.

3 (b). Furthermore, in their revision the author argues that the lack of clinical diagnosis could have affected the large GWAS by Doust et al. However, earlier GWAS that used both clinical diagnosis (Gialluisi et al 2021 https://www.nature.com/articles/s41380-020-00898-x) and quantitative measures (Gialluisi et al 2019) also failed to provide support for the genes selected here. Notably these GWAS analysed samples that led to the identification of some of the dyslexia linked regions.

Our selection of the regions and genes preceded the publication of the paper by Doust et al, 2022 and both papers by Gialluisi et al, and results from these papers do not change our strategy. Sequencing of samples was also in progress before any of these papers were published. We note, also, that 4 of the 5 genes that we looked at were among the subset of 8 genes that Gialluisi et al 2019 evaluated (DYX1C1, KIAA0319, DCDC2, GRIN2B and DNAAF1=DYX1C1), with similar reasons for gene selection to ours, and with both groups making this decision independently.

---

## [Editor Report · Decision Letter 3]

21 Apr 2025

Targeted analysis of dyslexia-associated regions on chromosomes 6, 12 and 15 in large multigenerational cohorts

PONE-D-23-41888R3

Dear Dr. Raskind,

We’re pleased to inform you that your manuscript has been judged scientifically suitable for publication and will be formally accepted for publication once it meets all outstanding technical requirements.

Kind regards,

Madelon van den Boer

Academic Editor

PLOS ONE

Additional Editor Comments (optional):

Thank you for your kind and thorough response in this final round of revisions. 

Reviewers' comments:

NA

---

## [Editor Report · Acceptance letter]

PONE-D-23-41888R3

PLOS ONE

Dear Dr. Raskind,

I'm pleased to inform you that your manuscript has been deemed suitable for publication in PLOS ONE. Congratulations! Your manuscript is now being handed over to our production team.

Kind regards,

on behalf of

Dr. Madelon van den Boer

Academic Editor

PLOS ONE